

**Changes in glacial lakes in the Poiqu River Basin in the central Himalayas**
Pengcheng Su [1, 2], Jingjing Liu *[1, 2], Yong Li [1, 2], Wei Liu [1, 2], Yang Wang [1], Chun Ma [1, 2]
1. Key Laboratory of Mountain Hazards and Surface Process / Institute of Mountain Hazards and Environment,
Chinese Academy of Science, Chengdu, 610041, China
2. University of Chinese Academy of Sciences, Beijing, 100049, China
**Abstract:** The Poiqu River Basin contains 162.2 km$^2$ of ice and 19.9 km$^2$ of glacial lakes. The
remote sensing data over the last 40 years have been used to identify 147 glacial lakes in the basin
and clearly revealed the retreat of glaciers and the growth of glacial lakes at accelerating rates, in
parallel to warming climate in the Himalayas. Based on remote sensing images and digital
elevation model (DEM) analysis, the area and water changes in glaciers and glacial lakes are
analyzed in detail, and a water balance equation (WBE) is proposed to account for the mechanism
of lake growth. The WBE includes water supplies from rainfall runoff, ice and snow ablation,
glacial retreat, and water losses due to infiltration and evaporation. As each water contribution
item specifically depends on local weather and morphology, the WBE provides a direct link
between glacier and glacial lake changes and climate changes under local conditions. Operation of
the WBE for five major glacial lakes in the Poiqu River Basin has revealed that water from
glaciers and snow cover dominates the growth of lakes. Lakes are found to vary in different ways
even with similar backgrounds, depending strongly on local weather and geomorphology
conditions. The WBE is not only applicable for predicting future changes in glacial lakes under
climate warming conditions but is also useful for assessing water resources from rivers in the
central Himalayas.
**Keywords:** glacier; glacial lake; global warming; water balance; Poiqu River Basin; Himalayas





## 1 Introduction

Worldwide glacial retreat due to global warming has led to great changes in alpine glacial
lakes (*IPCC, 2013; Mergili et al., 2013; Nie et al., 2014; Wang and Zhang, 2014; Prakash and*
*Nagarajan, 2017*). Apart from glaciers in the Arctic and Antarctic, accounting for 45.5% and
18.8% of the total, respectively, most glaciers are distributed in Asia, mainly in central, southeast
and southwest Asia, accounting for 13.8% of the total (*Mu et al., 2018*). Most glaciers retreat at
increasing rates (*Solomina et al., 2016*). In the mountains of the Andes, Caucasus, Altay, and the
Canadian Arctic region, glaciers have reduced in thickness by 3.6-11 m, while in the mountains of
Tianshan, Alaska, Svalbard, Alps, and the Pacific coast, glaciers have thinned by up to 30 m
(*Zhang, et al., 2015; 2019*). As the warming rate is much higher in Asian alpine areas, it is
expected that approximately 36% of the ice will be lost by the end of this century (*Kraaijenbrink*
*et al., 2017*).
The glacier inventory indicates that the area of glaciers in the Tibetan Plateau has reduced by
9.5% (767 $km^2$) in the last 40 years (*Wang et al., 2012; Nie et al., 2017*). The reduction in the
south is much larger than that in the north (*Wei et al., 2014*), and the greatest changes in glacier
area and length occur in the Himalayas (*Yao et al., 2012*). The retreat of glaciers in the Himalayas
has led to the expansion of existing glacial lakes and the generation of new lakes (*Richardson and*
*Reynolds, 2000; Komori, 2008; Bolch et al., 2008; Bajracharya et al., 2007; Yao, 2010; Shrestha*
*and Aryal, 2011; Raj et al., 2013*). Approximately 4950 lakes were identified in the Himalayas in
2015, mainly of which were located between altitudes of 4000 and 5700 m, with a total area of
455.3 ±72.7 $km^2$, which has increased by approximately 14.1% since 1990 (*Nie et al., 2017*). In
particular, in the central Chinese Himalayas, the glacial lake area has increased greatly, from
166.48 to 215.28 $km^2$, although the number of lakes has decreased, from 1750 to 1680 in the last
40 years (*Wang et al., 2012*). This implies that the changes in glacial lakes are mainly due to the
expansion of existing lakes. Statistics show that the expansion accounts for 67% of the area
increase, while the formation of a new glacial lake contributes only 33%. This expansion depends
on the fact that most lakes are fed by melt water of glaciers. In fact, the lakes associated with
glaciers increased by 122.1% in area during 1976-2010 in the central Himalayas, while lakes
without melt water remained steady, increasing only 2.8% in area during the same period (*Wang et*
*al., 2015*). Thus, the increase in glacial lakes is associated with the retreat of glaciers.



Glacial retreat appears most remarkably in the south central Himalayas (*Nie et al., 2017*),
where the last 30 years have witnessed a glacier length reduction of approximately 48.2 m on
average and area reduction at a rate of 0.57%. In the southern Himalayas lies the Koshi River,
which has attracted great attention because glaciers have decreased by approximately 19% in area
in the last 40 years (*Shangguan et al., 2014; Xiang et al., 2018*), and the melt rate has been
accelerating in the last decade. From 2000 to 2009, this glacial lake increased by 10% in area (0.7
$km^2$/a) (*Wang and Zhang, 2014*). Moreover, the Poiqu River (Bhote Koshi River), a tributary of
the Sun Koshi River, is a more active location for dramatic changes in glaciers and glacial lakes.
Landsat data indicate that the annual retreat rate of Poiqu was approximately 0.54% between
1976-2010, and in 1986-2001, it increased up to 1.3% per year (*Chen et al., 2007*) and has been
accelerating since 2000 (*Xiang et al., 2014*). Consequently, the glacial lake increased by 47% in
area (0.37 $km^2$/a) (*Chen et al., 2007*) in 1986-2001.
The retreat of glaciers and the growth of lakes are generally believed to be caused by rising
temperatures and decreasing rainfall (*Yao et al., 2012; Xiang et al., 2014; Mir et al., 2014*).
Records show that the temperature in the west Himalayas has increased by approximately 1.7°C in
the last century, while the rainfall is decreasing (e.g., *Bhutiyani et al., 2009; Mir et al., 2015a;*
*2015b*). In particular, observations in the Tibetan Plateau indicate that there is a strong tendency of
temperature rise at high elevations (*Liu and Chen., 2000*), and the rising rate increases with
elevation, reaching its highest at approximately 4800 to 6200 m. (*Qin et al., 2009*), which is in the
range of glacier development.
Although it is well acknowledged that glaciers and glacial lakes are sensitive indicators of
climate change, most studies are merely taken at large spatial and temporal scales, and only a
gross tendency is outlined for the changes (*Chen et al., 2007; Wang and Zhang, 2014; Wang et al.,*
*2015; Wang and Jiao, 2015; Xiang et al., 2018; Zhang et al., 2019*); special cases are only
concerned with lake breaks (*Xu and Feng, 1988; Chen et al., 2007; Wang et al., 2018; Nie Y et al.,*
*2018*). In the present study, we use multisource images from the last 30 years to explore the lake
variation in the Poiqu River Basin and provide a quantitative analysis of the water balance, which
leads to a method for assessing glacial lake change under a warming climate and sheds new light
on the mechanism of glacial lake evolution.



## 2 Study area

### 2.1 Geomorphology of the Poiqu River Basin

Remote sensing data and field surveys indicate that the Poiqu River Basin is an area of concentration for glaciers and glacial lakes. The Poiqu River (known as the Bhote Keshi River in Nepal) is the boundary river between China and Nepal, which is located along the southern slope of the central Himalayas, between the Himalayas and Laguigang Mountains.

Within the Chinese territory, the length of the Poiqu River is 90 km, and the basin area is $2.54 \times 10^3$ km², dropping from a high of 5810 m at the source peak to a low of 1750 m, with an average relief of 41‰. The section from Nyalam County to Zhangmu port is approximately 25.27 km in length, the average elevation difference is 2010 m, and the average vertical drop is 79.5‰. According to the ZY-3 satellite image on August 28, 2019, the total ice area in the Poiqu River Basin is approximately 162.2 km², and the total glacial lake area is 19.9 km² (Fig. 1).

**Fig. 1 Poiqu River Basin as a typical glacial lake in the central Himalayas**

Geologically, the Poiqu River Basin is located in the central Himalayan terrane, which was formed by the Indian-Eurasian plate collision. The Himalayan orogenic belt has a crystalline basement complex anticline north wing (the anticline is located in Nepal). The whole basin runs through the northern Himalayan Tethyan sedimentary rock belt, high Himalayan, low Himalayan and other tectonic units, all of which are bounded by the South Tibet detachment fault (STDS) and the main central fault (MCT) (Fig. 2).

**Fig. 2 Geological background of the Poiqu River Basin (Base map based on Pan, 2013)**

The Sun Koshi River developed and cut through the MCT, and the Poiqu River has experienced many tectonic movements since the Pliocene; however, the difference in the local zone due to tectonic effects has been relatively reduced because of the large uplift of the plateau. The uplifted mountains continue to be eroded and denuded, while the relatively sloped gullies receive uneven amounts of loose accumulation. Under such a background, the Poiqu River is mainly characterized by alluvial and diluvial valleys, with widths of 20 m to 200 m. The riverbed twists and turns and



develops multilevel terraces. To the south of Nyalam county, the valley bottom is narrow with
steep walls, most of which are V-shaped and Y-shaped valleys. The longitudinal section of the
riverbed is undulating, with multiple waterfalls and turbulence.
**2.2 Climate and hydrology**
The main Himalayas edge divides the Poiqu River into two climate zones: the northern zone,
featured by Yalai village, is temperate and subhumid, with an average annual temperature ($T_a$) of
3.5°C and rainfall ($R_a$) of 1100 mm; the southern zone, featured by Zhangmu town, is in the
subtropic monsoon climate, with $T_a$ of 10~20°C, $R_a$ of 2500~3000 mm, and frost-free period of
250 days, which is the area with the highest concentration of rainfall worldwide. According to
weather records in Zhangmu, $T_a$ is approximately 12°C, $R_a$ has been 2820 mm in recent years, and
more than 80% of rainfall occurs between June and September. Fig. 3 displays the 2016 daily
temperature records in the study area. The average temperature is similar in Nylamu and Quxiang,
where the positive temperature is concentrated between April and October, coincident with the
rainy season.
The Poiqu River has 5 major tributary rivers larger than 100 km$^2$, i.e., Chongduipu, Keyapu,
Rujiapu, Tongqu, and Dianchanggou. Rainstorms during the rainy season often cause floods in
these rivers. Field surveys indicate that the average annual discharge in the Chongduipu tributary
is 5.8 m$^3$/s, and it is 31.7 m$^3$/s in the Poiqu mainstream, with high seasonal fluctuations.
**Fig. 3. Monthly temperature and precipitation records in the study area**
**3 Identification of glaciers and glacial lakes**
**3.1 Data sources and image processing**
**3.1.1 Data sources**
Landform data are mainly from ALOS-12.5 m and ASTER-30 m elevation data, which are used
for correcting remote sensing data and interpretation. Geological data come from geological maps
of the Tibet Plateau. Remote sensing data come from the Landsat, GF-2, ZY-3, and UAV satellites,
as listed in Tables 1 and 2.
**Table 1 Data sources and features for interpretation of glaciers and glacial lakes**





**3.1.2 Image processing**
Generally, we use the fusion method to integrate the multispectrum data of 4 m GF-2 and the
full color data of 1 m GF-2 to create a base map for interpretation. In detail, for TM data, we use
742 band combinations and 432 combinations to highlight the colors of glaciers and glacial lakes;
for the data from GF-2, we combine the 321 bands of true color and the standard 432 bands of
false color images. Then, the ratios between different bands of the multispectrum data are used to
create images at different gray levels.
For glaciers, reflectivity is large for green light and small for intermediate infrared light. Thus,
the NDSI is employed to obtain the gray images, which is calculated as follows (Zhang *et al*.,

10    2006):

11          $NDSI = (float(b_{Green}) - float(b_{SWIR})) / (float(b_{Green}) + float(b_{SWIR}))$        (1)

where $B_{Green}$ is the green band and $B_{SWIR1}$ is the intermediate infrared band. The index falls
between -1 and 1, which can be further readjusted using ENVI software to provide the proper
threshold. In this study, we set NDSI>0.35 as the threshold for glaciers.
For glacial lakes, reflectivity of blue light is large and it approaches zero for near infrared, so
the NDWI is used to create the gray images, which is calculated as follows (Zhang *et al*., 2006):

17          $NDWI = (p(Green) - p(NIR)) / (p(Green) + p(NIR))$        (2)

where p(Green) and p(NIR) are the reflectivities of green and near infrared light, respectively.
Similar to the NDSI, we set NDWI>0 for water, which can be used as a criterion to identify glacial
lakes since there are no other water bodies in the study area.
**3.2 Identification of glaciers and glacial lakes**
Glaciers and glacial lakes present special shapes, colors, textures, and band combinations in the
images. Fig. 4 displays the images with characteristic marks, and Table 2 lists the signs for
identifying types of glaciers and glacial lakes. In practice, these elements are combined with
morphology and DEM data to delineate the boundary of lakes or glaciers. Moreover, moraines,
deposits, and colluvium are also identified by their marks and spectral features.
In particular, glaciers are located near mountain tops and limited to certain elevations. Glaciers
usually have tongue-shaped fronts with flow lines, and the uppermost boundary coincides with the
mountain edge, with ice cracks on the trailing edge, which are shown in black in the image.
Glacial lakes occur below glaciers, usually elliptical or flat, with smooth boundaries.



1       **Fig. 4. Characteristics of glaciers and glacial lakes in the Poiqu River Basin**

**Table 2 Interpretation signs for glaciers and glacial lakes(Six pictures are from Google**

**Earth images and two pictures are from GF-2 images. They are signed in the lower right**

**corner)**

**3.3 Results of interpretation**

A total of 147 glacial lakes and related glaciers have been identified in the Poiqu River Basin,

with a glacier area of 162.2 $km^2$ and a glacial lake area of 19.9 $km^2$. Table 3 lists the types and

numbers of each lakes. Most of these lakes are end moraine lakes.

**Table 3 Types of glacial lakes in the Poiqu River Basin**

These lakes have areas ranging between $1.66 \times 10^{-4} \sim 5.50$ $km^2$, and 125 lakes are smaller than

0.1 $km^2$. More than 60% of lakes are located at altitudes between 5000 ~ 5500 m. Lakes larger

than 0.1 $km^2$ are mainly in the tributaries of Keyapu, Rujiapu, and Chongduipu in upper Poiqu and

in Zhangzangbu in middle Poiqu. As listed in Table 4, more than half of the lake area is located in

Chonduipu, approximately 9.51 $km^2$, and the second largest is Keyapu at approximately 5.44 $km^2$.

These lakes account for 83% of the total area of glacial lakes. The table also lists the distance of

the lake to its connected glacier, indicating that most lakes are nearly linked to the glacier and thus

their changes are expected to be well correlated.

**Table 4 Typical glacial lakes in tributaries of the Poiqu River Basin**

Fig. 5 provides detailed distributions of glacial lakes in 4 major tributaries of Poiqu:

Chongduipu tributary (Fig. 5A), Zhangzangpu tributary (Fig. 5B), Keyapu tributary (Fig. 5C) and

Rujiapu tributary (Fig. 5D), where we have relatively large glacial lakes for consideration, i.e.,

Galongco Lake (5.50 $km^2$), Gangxico Lake (4.60 $km^2$), Jialongco Lake (0.60 $km^2$), Longmuqieco

Lake (0.52 $km^2$), and Cirenmaco Lake (0.33 $km^2$). The features of the tributaries are as follows:



**Fig. 5 Distribution of glaciers and glacial lakes in the major tributaries of the Poiqu River**
**Basin**
1) Chongduipu lies in the western part of middle Poiqu, with a long, lobate form and
U-shaped channel, which flows from northwest to southeast. Chongduipu has four tributaries, and
the largest glacial Lake Galongco is located in the Jirepu tributary and supplied by the Jipuchong
glacier on the southeastern slope of Mt. Shisha Pangma.
2) Zhangzangbu joins Poiqu from the east in the middle reach in the form of broad branches
and V-shaped channels, which deeply cut the valley and leaves flow marks of approximately 30 m.
Glaciers are mainly distributed in the upper reaches, and Cirenmaco Lake is located in a tributary
in the eastern source area.
3) Rujiapu is a tributary of Tongqu and thus a secondary tributary of Poiqu. Rujiapu lies in
the eastern part of the upper reaches, forming long branches and U-shaped channels. It has a
90°-turn near the mainstream, flowing from northeast to southwest, and the glacial lakes are
concentrated in the southeast. Moreover, the Rujiapi tributary has four tributaries with
distributions of glaciers and lakes.
4) Keyapu lies in the upper western part of Poiqu, near Chongduipu in the source area. Keyapu
has broad branches and a U-shaped channel. Glaciers and glacial lakes are mainly distributed in
the southeast.
Table 5 lists basic parameters of the tributaries, which are crucial for the formation and
evolution of the lakes, and parameters for the major lakes in the present state, based on
interpretation of 2018 images, are listed in Table 6.
**Table 5 Parameters of the glacial lake tributaries**
**Table 6 Basic parameters for major glacial lakes in the Poiqu River Basin**
**4 Changes in glaciers and glacial lakes**
**4.1 Variations in glaciers and glacial lakes**
Interpretations of the multisource images allow for detailed scrutiny of changes in glaciers
and glacial lakes. In this way, we obtain the areas of glaciers and glacial lakes in recent decades.





Fig. 6 shows the total changes in glacier and glacial lake areas in Poiqu since 1977, where the
dotted line means that the curve is inferred only because of the lack of data before 1999. Despite
the possible uncertainty before 1999, the gross tendency of glacier loss and glacial lake growth is
clear. The retreat rate of glacier area reaches 43.6%, which is approximately 2.98 km$^2$/a;
accordingly, the glacial lake area has expanded by 169%, which is approximately 0.19 km$^2$/a.
Since 2004, the retreat rate has reached as high as 7.2 km$^2$/a, while the growth rate of the lake has
reached 0.44 km$^2$/a (in Table 7).
This finding is comparable to the results from the literature. For example, from 1975 to 2010,
glaciers decreased by 19% in area (*Xiang et al., 2014*), while glacial lake area increased by 83%
(approximately 0.26 km$^2$/a) from 1976-2010 (*Wang et al., 2015*). In 1986-2001, the glacial area
increased by 47% (approximately 0.37 km$^2$/a) (*Chen et al., 2007*).
For comparison, glacial lakes increased by 29.7% in the entire Chinese Koshi River (including
Poiqu and six other tributary rivers) in 1976-2000 (*Shrestha and Aryal, 2011; Wang et al., 2012*) at
a rate of approximately 1.6 km$^2$/a. In the Koshi River, the glacier area has decreased by 19%
(approximately 23.48 km$^2$/a) (*Shangguan et al., 2014; Xiang et al., 2018*), and the glacial lake area
has increased by 10.6%. In 2000-2010, the glacial lake increased by 6% in area (approximately
0.72 km$^2$/a) (*Wang et al., 2015*). This result means that Poiqu undergoes more dramatic changes in
glaciers and glacial lakes. In particular, the Galongco and Gangxico Lakes have increased up to
500% and 107%, respectively, following area decreases in their connected glaciers by 40%.
**Table 7 Area variations and annual speeds of glaciers and glacial lakes in Poiqu river**

22                                    **Basin since 1977**

23              **Fig. 6 Area variations in glaciers and glacial lakes in Poiqu**

**Table 8 Area variations in 5 typical glacial lakes and their glaciers since 1977**
Figs. 7-10 show pictures for the five major lakes and their connected glaciers (or the so-called
"mother glaciers", because they are the sources of generation for the connected glacial lakes) in
different years between 1977 and 2018. It is easy to calculate the area of glaciers and glacial lakes
in each stage, as listed in Table 8. (The data sources for the images in different years are listed in



1    Table 1 and Table 2).

2        For more details, we construct the annual variation in the lakes from the historical data; Fig.

3    11 shows the variation in Galongco Lake since 1977, which increased abruptly from 1.77 to 5.50

4    km$^2$ between 1977 and 2018.

6        **Fig. 7 Comparison of area change between Cirenmaco Lake and its connected glacier**

8        **Fig.8 Comparison of area change between Gangpuco Lake and its connected glacier**

10       **Fig. 9 Comparison of area change between Ganxico Lake and its connected glacier**

12       **Fig. 10 Comparison of area change between Longmuqieco Lake and its connected glacier**

14       **Fig. 11 Variation in the area of Galongco Lake (1977-2019) (The left image is from © Google**

15       **Earth and the right image about the Galongco Lake is from UAV image)**

19       The retreat-growth correlation can be seen more clearly from the large lakes mentioned above,

as shown in Table 9 and Fig. 12. The gross tendency of glacial retreat and glacial lake growth is
also remarkable here. Notably, there was a sudden decrease in area in 1981, simply because there
was an outburst (*Xu and Feng, 1988*). Thus, historical anomalies in glacial lake areas may be
caused by lake outbursts.

**Table 9 Annual rates of change in 5 typical glacial lakes and their glaciers**

**Fig. 12 Retreat of 5 typical glaciers, growth of 5 typical glacial lakes and rates of change**

**in the Poiqu River Basin**

The five major lakes, Cirenmaco Lake, Galongco Lake, Gangxico Lake, Jialongco Lake, and

Longmuqieco Lake, have increased up to 30%, 74%, 40%, 200%, and 54% at rates of 0.01 km$^2$/a,



0.13 km$^2$/a, 0.07 km$^2$/a, 0.02 km$^2$/a, and 0.01 km$^2$/a, respectively, from 1977 to 2018.
Corresponding to the decrease in glaciers, the variations in glacial lakes under consideration
have presented three patterns in recent years:
1) Fluctuation in area, as in the case of Cirenmaco and Jialongco (Tables 8 and 9 and Fig.
12A).
Both lakes are located at relatively low altitudes (Jialongco is at 4306 m and Cirenmaco is at
4639 m), are sensitive to temperature and both experienced an outburst in this episode (in 1981
and 2002, respectively) and then increased steadily. Jialongco Lake even experienced a sudden
rise during 2006 and 2008 (Fig. 13), when the local temperature reached its 50-year peak.
Moreover, a field survey indicates that Jialongco has an overflow at 0.3 m$^3$/s in the rainy season,
meaning that the lake has reached its maximum and thus fluctuates, similar to ordinary lakes
undergoing seasonal changes. This finding implies that small amounts of variation in glacial lakes
do not mean that the related glaciers also vary by small amounts. Dramatic change in glaciers
results in a great loss of water but does not necessarily increase the size of the connected lake.
**Fig. 13 Rapid rise in Jialongco Lake due to glacial loss (2002-2009)**
2) Remarkable increase in area, as in the case of Galong Lake and Longmuqieco Lake
(Tables 8 and 9 and Fig. 12B).
Historic remote sensing data (1954 ~ 2018) indicate that Galongco formed in the late 1960s
as a result of a warming climate. Then, the lake increased steadily, with no marks of historic
outburst and no overflow events based on recent UAV images. Indeed, the lake level is still 10 m
below the front moraine bank, and it is only at 1 km downstream that the water flows from
infiltration. Thus, the lake has had little loss of water and increases steadily. Despite no field
survey data, the same case can be expected for Longmuqieco, which has similar altitude and water
supply areas and connected glaciers.
3) Gentle increase in area, as in the case of Gangxico (Tables 8 and 9 and Fig. 12C).
Gangxico is supplied by the back glacier. As the glacier is small, the lake grows slowly. Moreover,
Gongxico is hydraulically connected near Gongco and Galongco, and its water enters Gongco in
the southern area through infiltration, while the water of Gongco infiltrates into Galongco (Fig.
14). As Gongco has remained steady in last 50 years, Gongxico is also in a balanced state and





shows a small tendency to increase.
These observations suggest that glacial lakes change in various patterns even under the same
local conditions. Furthermore, little variation in glacial lake area does not necessarily mean that
there are no changes in related glaciers. In this sense, glaciers are more sensitive to changes in
weather or climate.
**Fig. 14 Hydraulically connected glacial lakes (Galongco, Gangco, and Gangxico)**
**4.2 Influences of temperature and precipitation**
As glaciers are sensitive to temperature, it is reasonable to consider the effects of weather on the
changes in glaciers and glacial lakes. Unfortunately, weather stations are very sparse in the
Himalayas, and no stations in the tributaries are under consideration; only records from nearby
stations are accessible. Near the study area, we have three weather stations in Nylamu, Quxiang,
and Zhangmu at altitudes of 3900 m, 3300 m, and 2200 m, which not only represent the vertical
variations in weather but also the variations from north to south. Chongpudui and Rujiapu are in
the northwestern and northeastern areas of Nylamu, respectively, and both rivers are similar to the
whole county in terms of weather conditions, so the temperature and precipitation for lakes (i.e.,
Galongco, Jialongco, and Longmuqieco) in these tributaries can be interpolated from the records
in Nylamu. Similarly, the weather of the Zhangzangbu (for Cirenmaco Lake) River is interpolated
from the records in Quxiang.
Combining the data from the three stations may comprehensively reflect the weather features
of the study area. The key factor for interpolation is the gradient of temperature ($R_T$) and
precipitation ($R_P$) varying with elevation. To obtain the $R_T$ and $R_P$, we take the records of Nylamu
and Zhangmu in 2016. The daily $R_T$ is defined as follows:
$$R_T = (T_N - T_Z) / (Al_N - Al_Z) \qquad (3)$$
where $T_N$ and $T_Z$ are the daily temperatures recorded in Nylamu and Zhangmu, respectively, and
$Al_N$ and $Al_Z$ are the altitudes of the two stations. This gives an $R_T$ of -6.1 ℃/km. As precipitation in
the study area is also governed by altitude, $R_P$ can be obtained in a similar way, i.e., the
precipitation difference divided by the altitude difference between the two stations, which gives a
value of -10 mm/km. The minus symbol means a decrease with altitude. Fig. 15 displays the

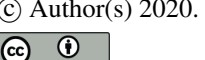



interpolated temperature and precipitation for the glacial lakes under consideration. Then, both the
interpolated temperature and precipitation for the target point can be obtained in the same way:
$$T_H = T_0 - R_T\Delta H, \text{ and } P_H = P_0 - R_P\Delta H \qquad (4)$$
where the subscript H means the altitude of the target points (i.e., the tributary rivers or the glacial
lakes) and 0 indicates the recorded values.

7                    **Fig. 15A. Interpolated cumulative temperature**

9                    **Fig. 15B. Interpolated annual precipitation**

11               **Fig. 15 Interpolated temperatures and precipitation for the glacial lakes**

Based on the interpolation, the temperature rises at a rate of approximately 0.02°C/a in Poiqu,
accompanied by a rainfall rate of 0.76 mm/a between 1989 and 2018.
Fig. 16 shows the temperature series in the last forty years in contrast to the areas of glaciers
and glacial lakes, indicating that the temperature is negatively and positively related to glaciers
and glacial lakes. Fig. 17 shows the precipitation series in contrast to the areas of glaciers and
glacial lakes, indicating that the tendency of precipitation is negatively associated with glaciers but
positively associated with glacial lakes. On the other hand, it is found that precipitation is well
correlated with temperature, with a correlation coefficient larger than 0.5. In short, the growth of
glacial lakes following the retreat of glaciers is governed by warming conditions.

22               **Fig. 16A Changes in the area of glaciers vs. temperature**

24               **Fig. 16B Changes in the area of glacial lakes vs. temperature**

26          **Fig. 16 Changes in the area of glaciers and glacial lakes vs. temperature in Poiqu**

28               **Fig. 17A Changes in the area of glaciers vs. precipitation**

30               **Fig. 17B Changes in the area of glacial lakes vs. precipitation**



2        **Fig. 17 Changes in the area of glaciers and glacial lakes vs. precipitation in Poiqu**

Despite the remarkable fluctuation in various episodes, the weather conditions present a gross
tendency in parallel to the retreat of glaciers and growth of glacial lakes. In fact, the temperature in
the Tibet plateau increased at a rate of approximately 0.3-0.4°C per ten years, nearly two times the
global rate. For the case of the present study, the lake area increases by approximately 40% at a
rate of 0.28 km$^2$/a, which is clearly higher than the other regions in the Himalayas (*Nie et al.,*
*2017*).
In the following section, we propose a procedure to calculate the water balance for typical
glacial lakes, illustrating the weather effects on the changes in glaciers and glacial lakes in
different ways.
**5 Water balance for glacial lakes**
**5.1 Volume of glacial lakes**
To understand changes in glacial lakes, it is necessary to find the changes in water volume in
the lakes. Then, we must find the lake volume from the area. This procedure can be done using
ArcGIS tools. First, we create the DEM of the lake bottom using images at the time the lake
formed and the following periods. Meanwhile, we interpret the annual water level from a series of
images, which record the evolution of the lake (cf. Fig. 11). In detail, we create irregular triangle
nets (ITNs) under the control of contour lines and obtain the DEM since the formation of the lake.
Then, the average elevation of the lake boundary (i.e., the water level) can be obtained for many
years. Finally, we compare the DEM derived from the water level and the DEM before lake
extension and obtain the variation in the water level with the water volume (Fig. 18). For example,
Galongco has the following relationship between lake volume and area: $V_{\text{lake}} = 5.0A^{1.72}$ ($10^6$ m$^3$),
where $A$ is the lake area (in units of km$^2$).

28        **Fig. 18 Terrain reconstruction of GB Lake below the water level**

**5.2 Water balance equation (WBE)**
The observations above indicate that the expansion of glacial lakes is well related to the retreat



of glaciers, which in turn relies on changes in temperature and precipitation (rainfall and snow) in
recent years. Then, it is possible to propose the following water balance equation (WBE) for a
glacial lake:
$$\Delta V = \Delta P + \Delta G - \Delta I - \Delta E \qquad (5)$$
where $V$, $P$, $G$, $I$, and $E$ are the water quantities of the glacial lake, the water supplies from
precipitation (rainfall and snow), glacier loss and ice-snow melting, and water loss through
infiltration and evaporation, respectively; $\Delta$ represents the annual increment.
In detail, the items in WBE are closely related to weather and geomorphologic conditions and
can only be determined empirically.
1) Water supplies from precipitation ($P_R$, $P_S$)
This involves rainfall and snowfall. The water supply from rainfall ($P_R$) is governed by the
hydrological process in the valley. For a given valley, the runoff depends on the rainfall process
(often featured by intensity $R$ and quantity $Q_R$), the drainage area contributing to the lake ($S$), and
the geomorphologic factors such as slope $\theta$, vegetation cover, and permeability $K$. In general, this
can be expressed as follows:
$$P_R = f(R, Q_R, S, \theta, K) \qquad (6)$$
Water supplies from snowfall ($P_S$) also depend on temperature $T$, solar radiation $I_R$, snow
density $\rho_S$, and snow permeability $k$, in addition to the geomorphologic factors:
$$P_S = f(T, I_R, \rho_S, k, S, \theta, K) \qquad (7)$$
Then, the water supplied from precipitation is as follows:
$$P = P_R + P_S \qquad (8)$$
2) Water supplies from glaciers ($G$)
The major controlling factors are temperature $T$, solar radiation $I_R$, glacier density $\rho_G$, fracture
density $\sigma$, and geomorphologic factors:
$$G = f(T, I, \rho_G, \sigma, S, \theta, K) \qquad (9)$$
3) Water loss from infiltration ($I$)
Infiltration mainly depends on the permeability of the materials constituting the lake, and in the
present case, the materials are mainly moraines, which are generally poorly graded in terms of
grain composition and have high porosity. Infiltration also occurs underground and depends on the
substrate sediment of the valley channel downstream of the lake.
$$I = f(K, GSD, J) \qquad (10)$$
where $GSD$ describes the granular features of moraines and sediments (*Li et al., 2013, 2017*) in
terms of grain size distribution, and $J$ is the hydraulic slope between the water level and seepage



points.

In addition, when the lake is "saturated", i.e., the capacity reaches the maximum due to the limitation of the local landform, the lake will not increase in area, and the water supplies exceeding the capacity will be lost through overflow. In such a case, the supply is balanced by the loss.

4) Water loss from evaporation ($E$)

Theoretically, evaporation is controlled by temperature, solar radiation, lake area $A$, wind speed $v$, surface saturated vapor pressure $p$, and turbulent energy $\varepsilon$ (Lu *et al*., 2017):

$$E = f(T, I_R, v, p, \varepsilon) \tag{11}$$

However, for the present case, the effect due to evaporation is much smaller and is usually ignorable compared with the other contributing terms.

**5.3 Practical operation of the balance equation**

In practice, each item introduced in the WBE can be empirically estimated, especially in the present case, where we suffer from a severe lack of basic solar radiation and local weather data. In the following section, we provide a practical routine for the calculations.

1) Water supplies from rainfall and snow

In principle, the supply is equal to the runoff drainage to the lake, which is calculated using the standard hydrologic method for each rainfall event, depending on the temporal process and spatial distribution of the rainfall over the drainage area. However, for the case of glacial lakes, we have only annual area variation and weather data from nearby stations, and it is impossible to perform standard hydrograph calculations; instead, we reduce the calculation to the runoff of the slope (*Gao et al., 2019*):

$$P_R = \alpha S R_a \tag{12}$$

where $P_R$ is the runoff and employed here as the water supply from rainfall, $S$ is the drainage area contributing to the lake, $R_a$ is the annual rainfall, and $\alpha$ is the coefficient, depending on local conditions of the drainage slope, such as the material properties and vegetation cover, which is empirically determined as follows (Liang *et al*. 2018):

$$\alpha = 0.065 + 0.0086\theta + 0.33\text{ALs} \tag{13}$$

where $\theta$ is the slope angle, and ALs varies among arid, semiarid, semihumid, and humid areas. As the Poiqu River Basin is located in the semiarid area but has sufficient moisture content in air, ALs can be taken as the upper limit of 0.75. Then, $\alpha$ is mainly governed by the slope gradient of the drainage area to the lake.

2) Melt water from ice and snow melt

There have been various methods used in glacial hydrology (*Braithwaite and Olesen, 1989*). Physical models have incorporated many influencing factors, such as temperature and radiation intensity; thus, these models have high calculation accuracy. However, they do not apply to areas



lacking a sufficient database, as in the case in the Himalayas. Instead, empirical methods are
widely employed, among which the Degree-Day Model (DDM) is generally most used to calculate
the melting of glaciers and snow cover (*Kayastha et al., 2005; Zhang et al., 2006;*
*Pradhananga et al., 2014*). The DDM is practical, simple and well-accepted, considering the
influence of the degree-day factor (DDF) and the normal accumulated temperature. Following the
method, the melted thickness of the glacier (M) is determined by the production of DDF and the
positive cumulative temperature in a certain period (PDD, in units of d·°C):
$$M = DDF·PDD \qquad (14)$$
where DDF is in units of mm·d$^{-1}$·°C$^{-1}$, and varies with elevation (*Liu et al., 2014*). PDD can be
directly calculated from the daily temperature record, i.e., the cumulative temperature of the days
with temperatures higher than 2°C. In fact, the PDD involves two components applied to the melt
of snow cover and glaciers, PDD$_S$ and PDD$_G$. In other words, only the residual cumulative
temperature PDD$_G$ applies to glacial melting.
Then, the melt water quantity is the production of $M$ and the glacier area ($A_G$):
$$G = M·A_G \qquad (15)$$
Similarly, this also applies to the water supply from snow cover melting. DDF is generally hard
to obtain, but in Poiqu, we may make a reference to the results in the nearby area, 80 km away at
Mt. Everest. According to previous studies, the DDF is 16.9 for the Kunbu glacier at an altitude of
5350 m (86°52′E,27°59′N) (*Kayastha et al., 2005*), and the DDF is 8.21 for the Rongbu glacier at
the same altitude (*Liu et al., 2014*). Then, we take the average value, 12.6, as the overall DDF for
glaciers in Poiqu, and for individuals, we make some corrections depending on the slope
orientations of the glaciers. For the west-oriented slope (e.g., Cirenmaco Lake), the melt is
relatively more intense than the east-oriented slope (e.g., the Galongco and Gangxico Lakes); for
the cases of Jialongco and Longmuqieco, the slopes are north-oriented, the sunshine is shielded,
and the melt is relatively weak. Based on these results, we obtain a corrected DDF for each glacier
(Table 9).
According to studies on the snow cover of the Dokriani Glacier in the Indian Himalayas
(78°50′E, 28°50′N) (*Singh et al., 2000*), the DDF for snow is approximately 30% less than that for
glaciers. As this is geographically similar to the Poiqu area, a reduction rate of 30% can be used
for determining the DDF of snow cover for the glaciers and glacial lakes under consideration, as
listed in Table 6.
On the other hand, not all meltwater can reach the connected lake; some infiltrates into the
bed through the crevasses. This creates a loss of water supplies from melt water, and a reduction
coefficient, $R_c$, is considered when the water supplies are estimated (cf. Table 5).
3) Evaporation
The Poiqu River is located at high altitude, where the stored water is in a liquid state only in
July and August. It is reasonable to assume that the evaporation is very weak in this area and can
be ignored in the estimation of water balance. For this estimation, we take Gongco Lake as the





reference. The lake is located in the tributary of Chongduipu, similar to Galongco and Gangxico,
and at similar altitudes (5173, 5075 and 5218 m, respectively). However, it is distinctive in that the
Gangco does not receive a water supply from glaciers; the major water supplies come from rainfall.
Notably, Gongco has not increased in area, remaining at approximately 2.1 km$^2$ in recent years. It
is possible that the water supplies are balanced by the water losses due to infiltration and
evaporation. Since Gongco receives seepage flow from Gangxico and simultaneously feeds
Galongco through seepage, the supplies from rainfall can be considered balanced by evaporation.
However, according to the estimation, water supplies from rainfall are generally very small
compared with those from the meltwater of glaciers and snow cover. Therefore, evaporation is
negligible in the Poiqu River.
4) Infiltration

12        Water loss due to infiltration is controlled by the permeability of the moraine bank of the lake

and the sediment in the valley channel. As it is inaccessible to most glacial lake areas, we can only
trace the marks of infiltration through remote sensing images (including UAV and Google Earth)
(cf. the case of Galongco in Fig. 19).

16        For the permeability coefficient $K$, we conducted experiments on material samples from the

moraines and sediments, and it was found that $K$ (cm/s) is well related to the grain size distribution
(GSD) of the loose granular materials:

19                 $K=0.003D_c^{1.5}-29.46\mu^{2.5}-0.0196$ (R$^2$=0.9892)                 (16)

where $D_c$ and $\mu$ are GSD parameters (Li *et al.*, 2013; 2017), which can be directly obtained
from the granulometric analysis of moraine and sediment samples for each lake. Then, the
infiltration discharge can be calculated by Darcy's law:

23                 $Q = KJA$                 (17)

where $J$ is the hydraulic slope and $A$ is the infiltration area. For a given valley, the water loss
from infiltration is $I = QT$, with $T$ as the effective time for infiltration, which is mainly the rainy
season when the valley has flow water.
Based on the discussions above, we obtain a working list of parameters for calculating the WBE
(Table 10).

**Table 10 Parameters for the water balance calculation of glacial lakes**

**5.4 Calculations**
**5.4.1 Exemplification of Galongco**

Now, we apply the WBE to the five major lakes to see how the area has increased in recent

decades. For this procedure, we first take glacial Lake Galongco in 2006 as an example to show
the calculation process.

1) Geomorphologic background and related parameters

As mentioned above, Galongco Lake is located in a small tributary of Chongduipu, at an

altitude of 5075 m, in an area 5.5 km$^2$, and the drainage area to the lake, including slopes around



the lake, is 22.33 km$^2$. Two glaciers are directly connected to the lake in the northwestern and western parts of the upstream area, with a total area of 13.5 km$^2$ according to the GF-2 satellite images in 2018.

In 2006, the lake area was 3.93 km$^2$, and the glacier area was 13.06 km$^2$ (Fig. 19). Based on the DEM, the angle of the draining slope is estimated to be 23.7 ° on average, and thus, the runoff coefficient is 0.56 according to Eq. (13).

Following the background of the lake and glaciers, the DDFs for glaciers and snow cover are 12.6 and 8.3, respectively, and the reduction coefficients for glaciers and snow cover are 0.61 and 0.56, respectively (cf. Table 9).

**Fig. 19 Galongco Lake and the connected glaciers in 2006**

2) Weather conditions

The weather conditions are interpolated from the records in Nylamu; the annual temperature and precipitation in 2006 are shown in Fig. 20 according to this interpolation.

**Fig. 20 Temperature and precipitation of Galongco Lake in 2006**

Following the instruction above, the rainfall and snowfall in 2006 were 1.5 mm and 1545 mm, respectively, and the cumulative temperature was 282.3°C. Based on the DDM, the cumulative temperature for snow cover melt is 128.3°C, and thus the cumulative temperature for glacial melt is 153°C.

3) Infiltration

According to samples of moraine materials in the lake tributary, the GSD parameter $\mu$ is 0.03 and $D_C$ is 11.2 mm, which yields a permeability coefficient $K$ of 0.088 cm/s. According to Google Earth images, the infiltration area is approximately 8426 m$^2$, and the hydraulic slope is 0.13, which gives a discharge of infiltration of 0.96 m$^3$/s. Considering that only July and August have positive temperatures higher than 2°C, infiltration only occurs in these months.

4) Water supplies and losses

Based on the parameters described above and using formulas (6)-(9), we obtain the water supplies and losses:

(i) the water supply from rainfall ($P_R$) is 1.56×10$^5$ m$^3$;

(ii) the water supply from glacial melting ($P_S$) is 1.90×10$^6$ m$^3$;

(iii) the water supply from snow melting ($G$) is 8.11×10$^6$ m$^3$; and





(iv) the water loss from infiltration is ($I$)$5.92\times10^6$ m$^3$.
Therefore, the WBE provides a water supply of $4.25\times10^6$ m$^3$ to the lake in 2006, which
accounts for the area increase of 0.33 km$^2$.
In the same way, we can calculate the water balance for other years. Notably, for some years,
no data are available for glaciers or lakes (e.g., only three sets of data are available between 1988
and 2004); for these situations, we use an extrapolation method. Considering that the changes in
glaciers and glacial lakes have steady near-linear tendencies in recent years, we can assume that
both glaciers and glacial lakes in the years between 1988 and 2004 vary linearly, with the average
rate determined by the slope of the line linking the points of 1988 and 2004. Thus, we can infer the
area of glaciers and glacial lakes in those years. Specifically, for Galongco, the variation rate of
glaciers between 2004 and 2018 is -0.36 ($R^2$=0.8956), and the variation rate of glacial lakes is 0.15
($R^2$=0.8779), which provides a baseline for extrapolation in recent years.
Using the methods above, we obtain the water balance for Galomngco between 1988 and 2018, as
listed in Table 10, and the symbols in Table 11 are listed as follows:
$T_c$ – cumulative temperature;
$T_{cG}$ – cumulative temperature for glacial melting;
$T_{cS}$ – cumulative temperature for snow melting, which is $T_c$ - $T_{cG}$;
$M_G$ –melt thickness of a glacier;
$W_G$ – water supply from glaciers;
$W_{snow}$ – water supply from snow cover; and
$W_{total}$ –total quantity of water supplies.
**Table 11 Water balance for Galongco Lake between 1988 and 2018**
**5.4.2 Water balance for typical lakes**
Similarly, we can perform balance calculations for other lakes, from which we obtain the
variation in water quantity for the lakes since 1988 using the parameters listed in Table 6. Table 12
displays the comparison between the calculated water quantity and the observed quantity for the
five selected lakes.
**Table 12 Comparison between the calculated water quantity and the observed quantity**



The calculations generally agree with the observations, but it is noted that great discrepancy
occurs in the case of Jialongco, the lowest lake among the five samples at an altitude of 4306 m,
which experienced an outburst in 2002 and sudden rise during 2006 and 2008 due to dramatic
changes in the connected glacier (cf. Fig. 13). As the WBE does not consider the glacial dynamics
and dramatic changes in local conditions, the calculation cannot incorporate the sudden changes.
This means that the WBE operation should be further improved to incorporate the water variations
due to catastrophic processes.
However, the gross agreement between the calculation and observation does suggest that the
WBE has provided a practical and functional framework for understanding the characteristics of
changes in individual glacial lakes. Moreover, it provides a practical method for quantitatively
assessing the growth of glacial lakes. In particular, the calculation reveals that the lakes in Poiqu
have undergone different water supply balance proportions, which makes it possible to distinguish
among the local conditions of the lakes.
Table 13 lists the average fraction of water supplies from glaciers, snow, and rainfall over the
calculation period. It is obvious that lakes at relatively low elevations (i.e., below approximately
5000 m) are mainly supplied by glaciers, and lakes at high elevations are mainly supplied by
snowfall. For all these lakes, the water supplies from rainfall are much smaller, even below 5%,
and this can almost be ignored considering the accuracy of the estimation. This clearly reflects the
altitude effect on glaciers. At low altitudes, the cumulative annual temperature is positive and
directly melts the glaciers. At high altitudes, glaciers are covered by snow, and the positive
temperature mainly acts on snow cover. Indeed, several years have shown near-zero cumulative
temperatures for Gangxico Lake and Longmuqieco Lake, which results in a small fraction of
glacial ablation.
**Table 13 Fractions of various water supplies to the lakes**
Then, the WBE not only provides a method to account for the water supplies to glacial lakes
but also reveals differences between lakes. Although glaciers are sensitive to temperature, the lake
grows in various ways depending on local conditions, especially altitude and basin circumstances
(e.g., morphology and moraine materials).
**6 Discussions**
Based on the present study, we can remark on some of the problems concerning changes in





glaciers and glacial lakes under warming conditions.
1) Changes in temperature and precipitation have been recognized as effecting ice and snow
melt and leading to serious consequences for both nature and society (Immerzeel *et al*., 2012;
Immerzeel *et al*., 2013). Recently, the estimation of ice thickness distribution indicated that the
present-day glacier area in highly mountainous Asia will decrease by half at an accelerating rate,
approximately one decade ahead of schedule, as suggested in previous studies (Farinotti *et al*.,
2019). A detailed analysis in the present study proves that changes in glaciers and glacial lakes in
Poiqu are at remarkably high levels compared with other regions in the Himalayas (Nie *et al*.,
2017). In particular, although the glaciers are generally in their retreat phase, water supplies from
glaciers are still dominant in the central Himalayas. However, it is also noted that the fluctuation
of temperature and precipitation in local areas does not present a clear-cut tendency in parallel
with the retreat of glaciers or growth of glacial lakes. Changes in individual glaciers and glacial
lakes are dominated by local conditions but not global changes.
2) Mass balance for glaciers and ice caps is of great importance in Earth's hydrological cycle
and response to climate change (*Aizen and Aizen, 1997; Haeberli et al., 1999; Valentina Radić and*
*Hock, 2013; Lambrecht and Mayer, 2009; Huss, 2011; Huss and Hock, 2018*). The results of this
study provide a detailed scenario of water balance for individual lakes through operation of WBE
for typical glacial lakes, revealing details in water supplies from precipitation, glaciers, and snow
cover and water losses from infiltration. The WBE provides the mechanism for lake growth and
agrees well with the observations and image interpretations, and the calculation for individual
lakes has made up for the deficiencies in previous studies, which only gave an overall view of lake
expansion at the regional scale (e.g., Nie *et al*., 2017). In addition, the WBE operation has also
discovered that glacial lakes under similar background conditions may vary in different ways,
depending on local elements at small scales, which would be inevitably neglected in studies at
large scales. The lake may remain at their greatest sizes (e.g., at the maximal area of extension)
even if the glaciers undergo dramatic changes.
3) Furthermore, WBE operation is crucial to gain a better understanding of water supplies for
glacierized river basins. Near the study area, there are many rivers originating in the high Asian
mountains, such as the rivers of Yarlong Zangbo (Brahmaputra), Indus, Ganges, Nujiang (Salween)
and Lancangjiang (Mekong), but the quantification of water sources is usually highly uncertain



because of a lack of understanding of the hydrological regimes and runoff calculations (Winiger *et*
*al*., 2005; Bookhagen and Burbank, 2010; Immerzeel and Bierkens, 2012; Miller *et al*., 2012; Lutz
*et al*., 2014; Hassan *et al*., 2017). The proposed WBE calculation has revealed the variety of water
supplies from glaciers, snow cover, and precipitation for individual glacial lakes; thus, this
calculation is expected to be applicable for estimating glaciohydrologic processes in large
glacierized rivers.
4) Admittedly, the WBE for glacial lakes is proposed here only at the annual scale, which makes
it difficult to be accurate when considering individual lakes during a given period. This is mainly
due to the lack of data and ignorance of specific water supply and loss processes. For example,
runoff should be calculated for the tributary watershed using records for individual rainfall events,
which strongly depend on the watershed conditions (i.e., conditions of slope, channel, vegetation,
and soils or sediments, especially moraines for the lakes) and the rainfall pattern. However, in the
study area, and even in the Himalayas, only annual (and usually incomplete) weather records are
available at several points, and it is only possible to provide a gross estimate of the runoff simply
by the production of rainfall and watershed area. Similarly, water quantities from other sources
can only be best estimated for accuracy in terms of order of magnitude.
On the other hand, the WBE does not consider the dynamical processes of glaciers (Copland
*et al*., 2011; Dowdeswell *et al*., 1995), such as glacial surging, its hydrologic consequences or the
possible dramatic changes in morphology, such as the collapse of lakes or other surface processes
(e.g., icefalls, landslides, or debris flows due to earthquakes or extreme weather events), which
may bring dramatic changes that overwhelm the steady, gentle changes that occur over tens or
even hundreds of years. Therefore, the model cannot explain the sudden changes in glaciers and
glacial lakes, as in the case of Jialongco. In addition, the parameters involved for these items are
highly uncertain in practice, and systematic and detailed scrutinization is required to improve the
accuracy of the operation.
**7 Conclusion**
This study employed multisource images and identified 147 glacial lakes in the Poiqu River in
the central Himalayas and explored the detailed changes in major glacial lakes. Tracing the
evolutions of glaciers and glacial lakes over the last 40 years, we find that the glaciers have
undergone increasing retreat while the glacial lakes grew and expanded. The major lakes have

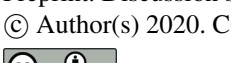



increased by up to 30% ~ 200% in area, at rates between 0.01 km$^2$/a and 0.13 km$^2$/a, which make
the Poiqu River an area of high levels of glacier and glacial lake changes in recent decades.
Detailed analysis of individual glacial lakes indicates that the lake grows in various patterns,
depending on local conditions of weather and geomorphology, or even occasional dramatic events
such as a lake outburst, icefall, or glacial surging. As these events are always inaccessible and
usually cannot be identified from images, abnormalities in glacial lake growth may provide hints
for those catastrophic occurrences. Meanwhile, small variations in lakes do not necessarily imply
no changes in glaciers and lakes.
Based on the changes in glacial lake area and DEM analysis, we abstracted the water change in the
lakes and proposed a WBE that governs the growth of the lake. As each item of the water
contribution specifically depends on local weather and morphology, the balance equation provides
a direct link between glacier and glacial lake changes and climate changes under local conditions.
Operation of the WBE for the five major glacial lakes in the tributaries of Poiqu River has
shown that individual lakes vary in different ways and receive water supplies from glaciers, snow
cover, and precipitation in different fractions. The results clearly reveal the altitude effect on
changes in glaciers and glacial lakes. At low altitudes, temperature is more effective for glacier
ablation, and lakes are mainly supplied by melted water from glaciers. At high altitudes,
temperature acts more on snow cover, and melted snow becomes the major water supply to lakes.
The difference between water supplies from glaciers and snow cover is as high as 50%, according
to the present cases. This implies that it is insufficient to apply weather or climate conditions to
individual glacial lakes at a large scale to determine climate effects on glacial lake changes.
**Acknowledgements**
This research is supported by the China Geological Survey projects (Grant No.
DD20190637), National Natural Science Foundation of China (Grant No. 41877261), National
Key Research and Development Plan of China (Grant No. 2017YFC1502502), the Strategic
Priority Research Program of the Chinese Academy of Sciences (Grant No. XDA23090202), the
Strategic Program of the Institute of Mountain Hazards and Environment, CAS (Grant No.
SDS-135-1701), and the CAS Key Technology Talent Program.





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



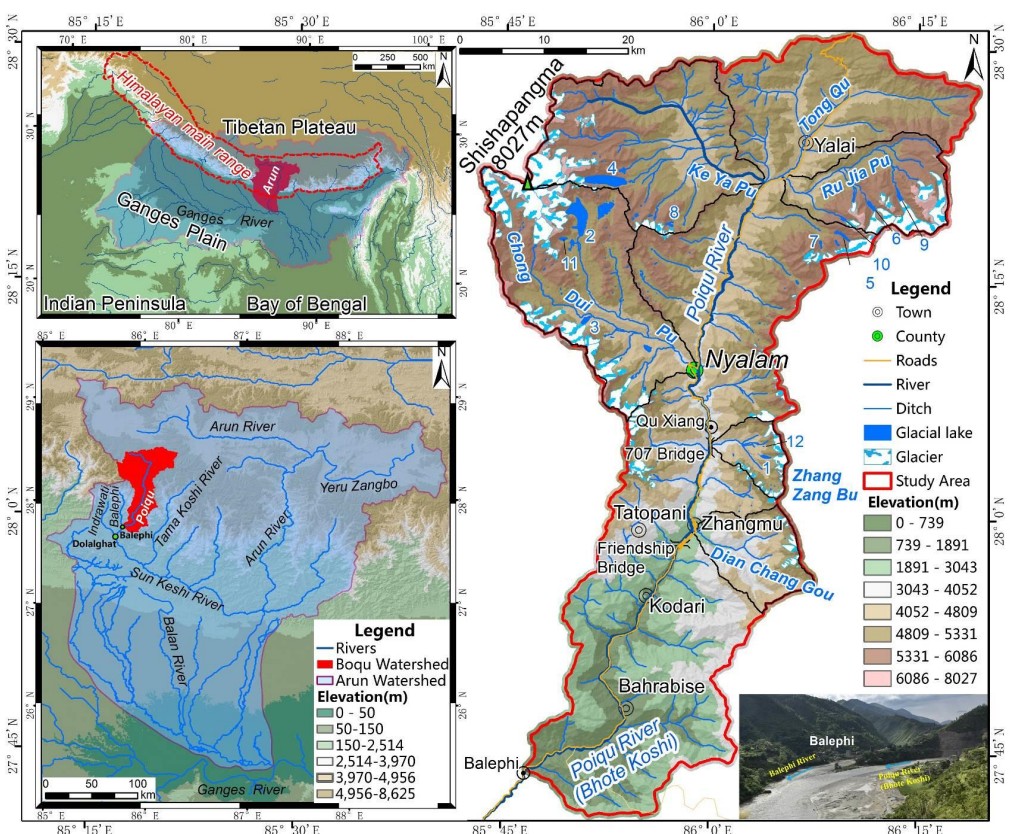

Fig. 1 Poiqu River Basin as a typical glacial lake in the central Himalayas





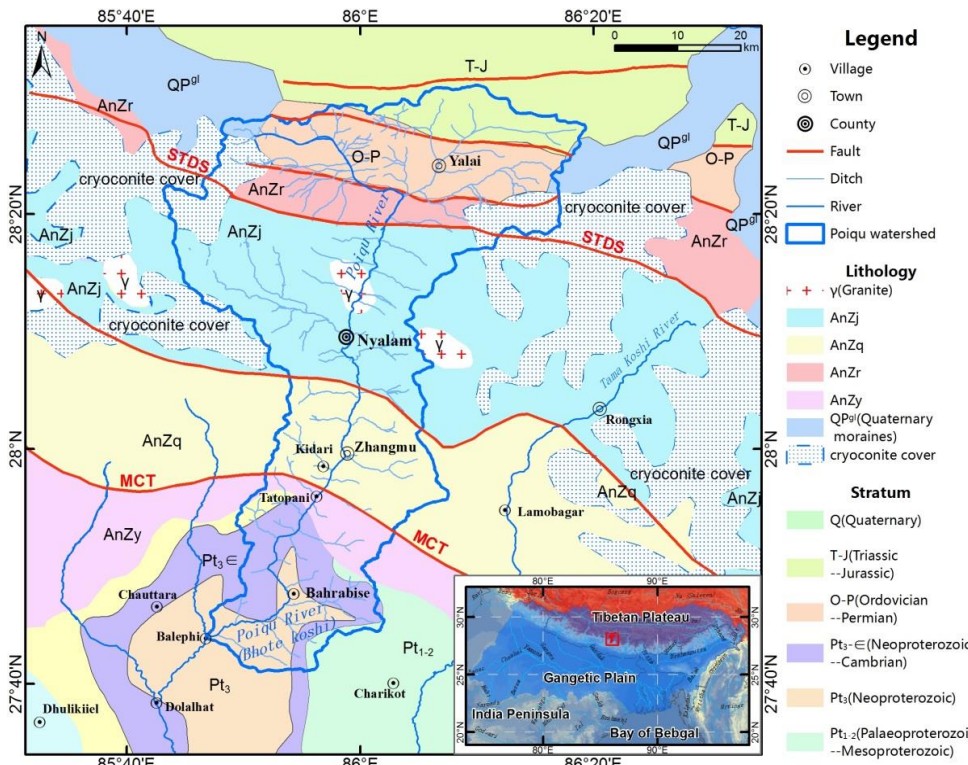

**Fig. 2 Geological background of the Poiqu River Basin (Base map based on Pan, 2013)**



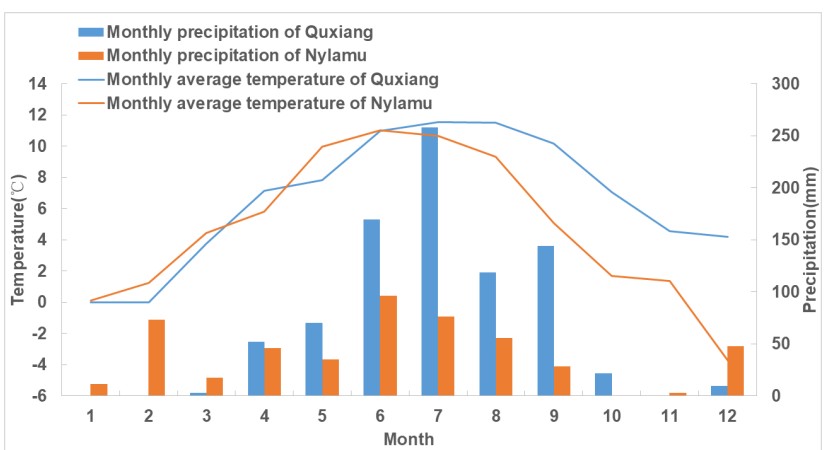

**Fig. 3. Monthly temperature and precipitation records in the study area**

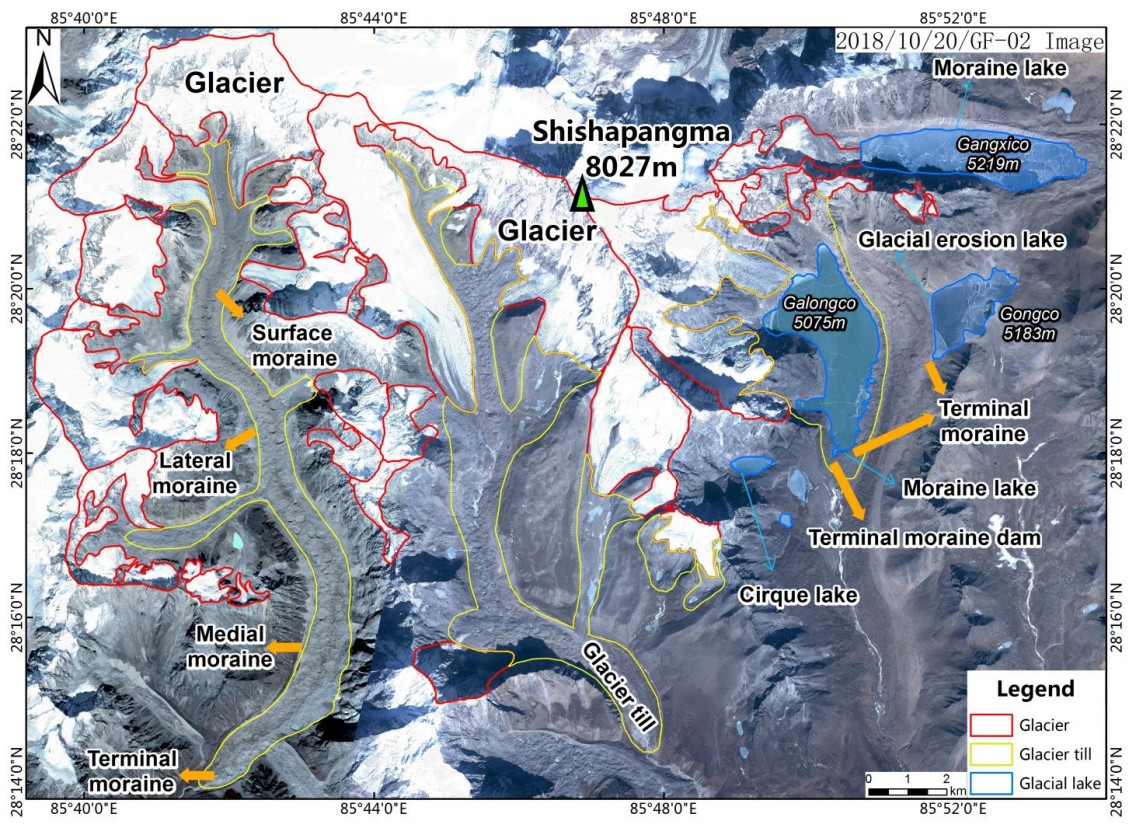

**Fig. 4. Characteristics of glaciers and glacial lakes in the Poiqu River Basin**

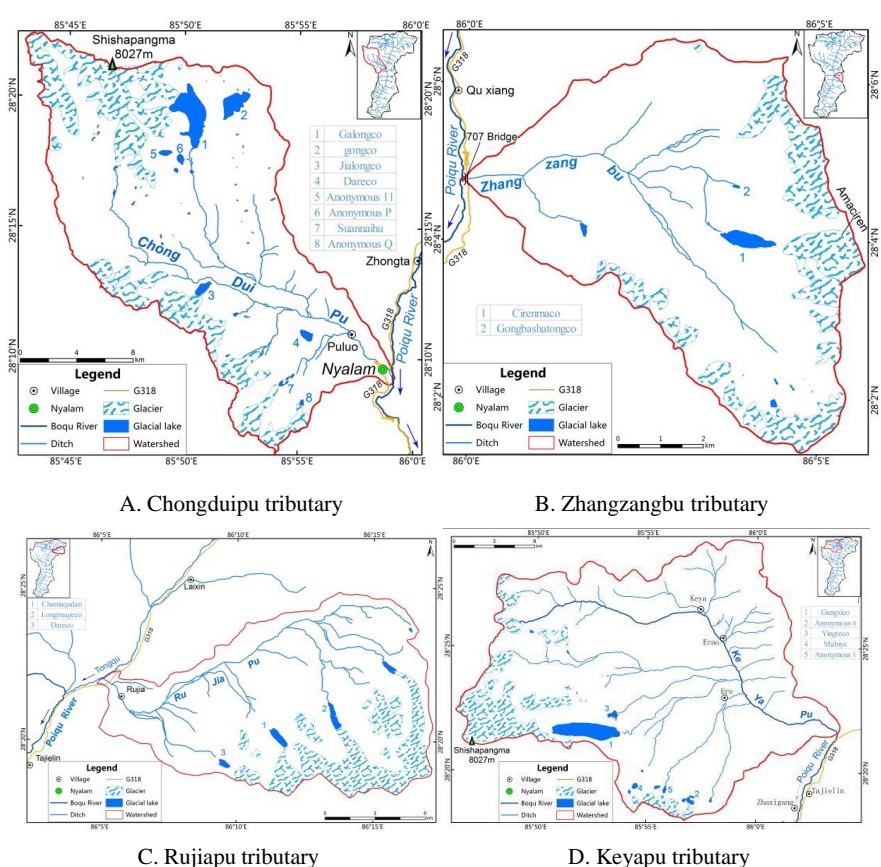

A. Chongduipu tributary                    B. Zhangzangbu tributary

C. Rujiapu tributary                    D. Keyapu tributary

**Fig. 5 Distribution of glaciers and glacial lakes in the major tributaries of the Poiqu River Basin**

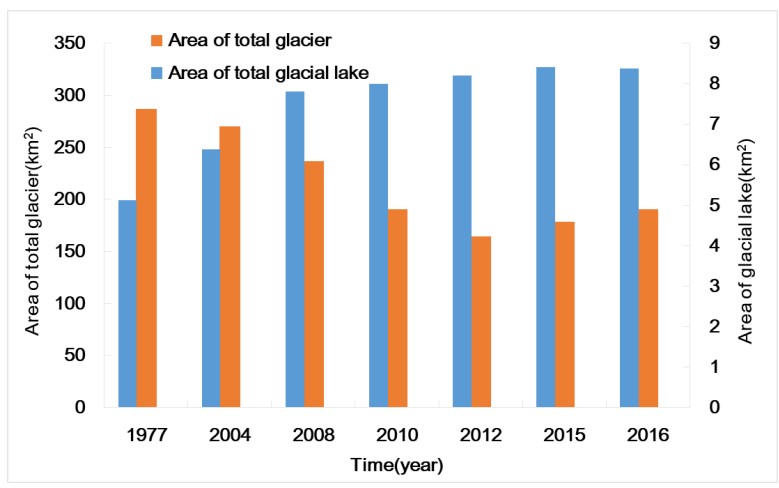

**Fig. 6 Area variations in glaciers and glacial lakes in Poiqu**



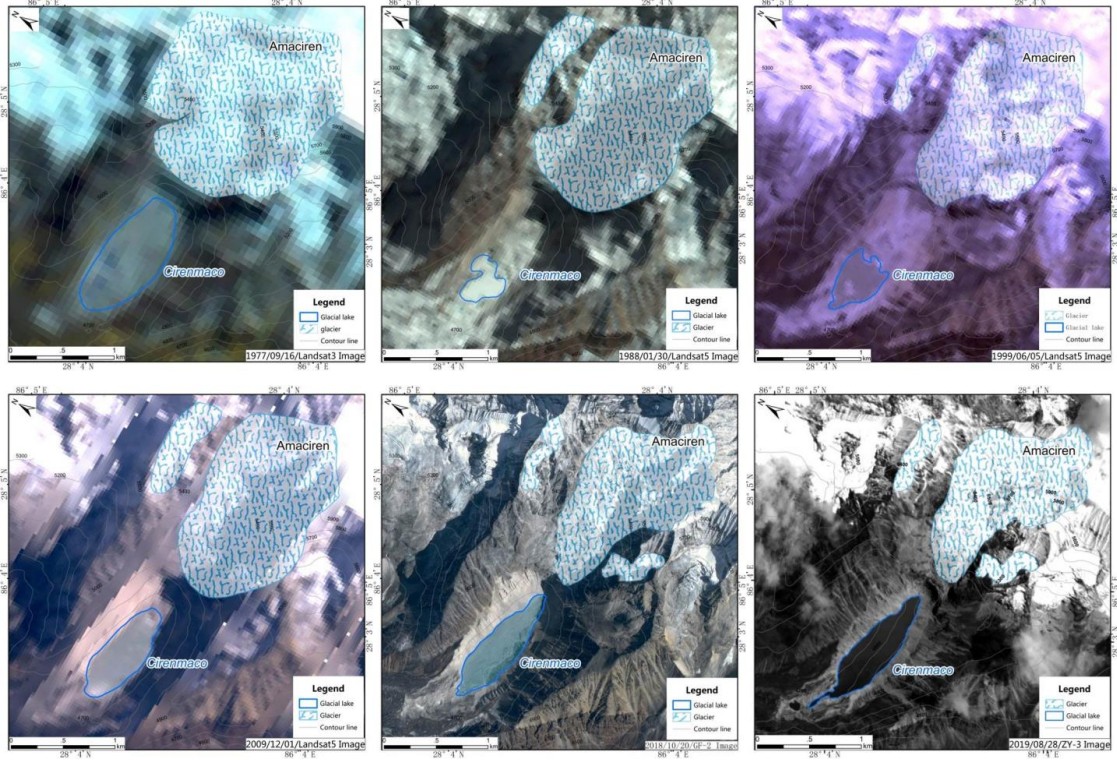

**Fig. 7 Comparison of area change between Cirenmaco Lake and its connected glacier**



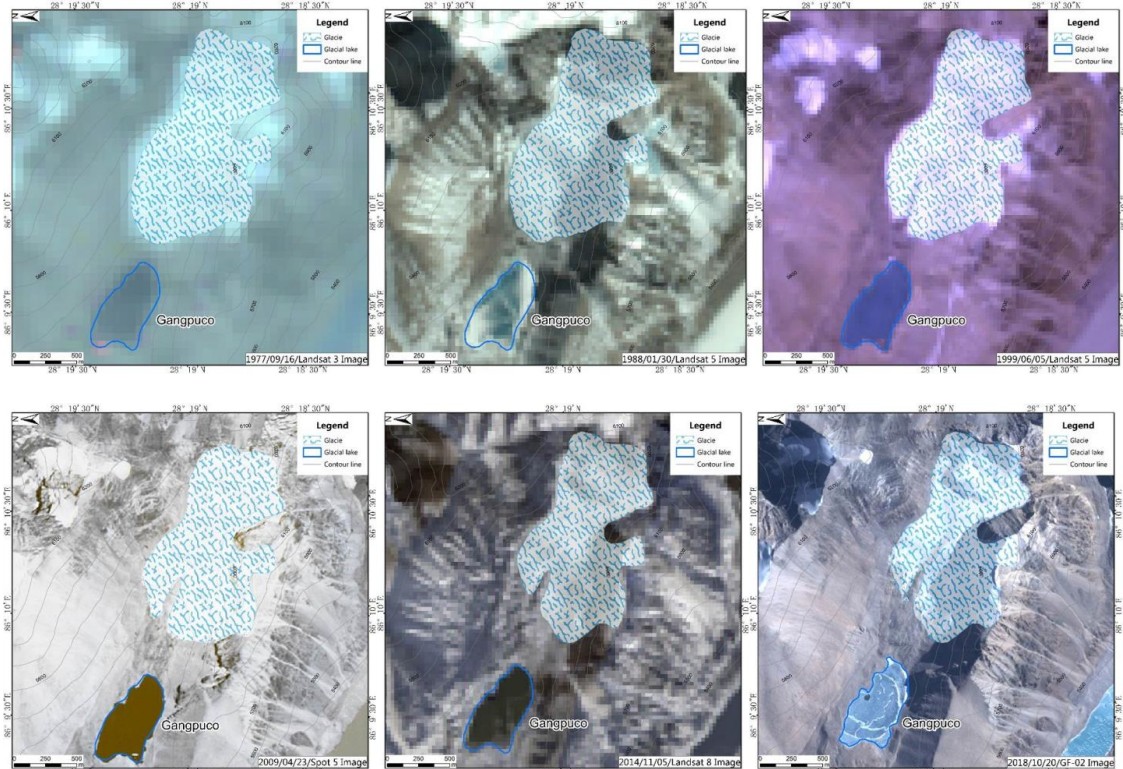

**Fig.8 Comparison of area change between Gangpuco Lake and its connected glacier**



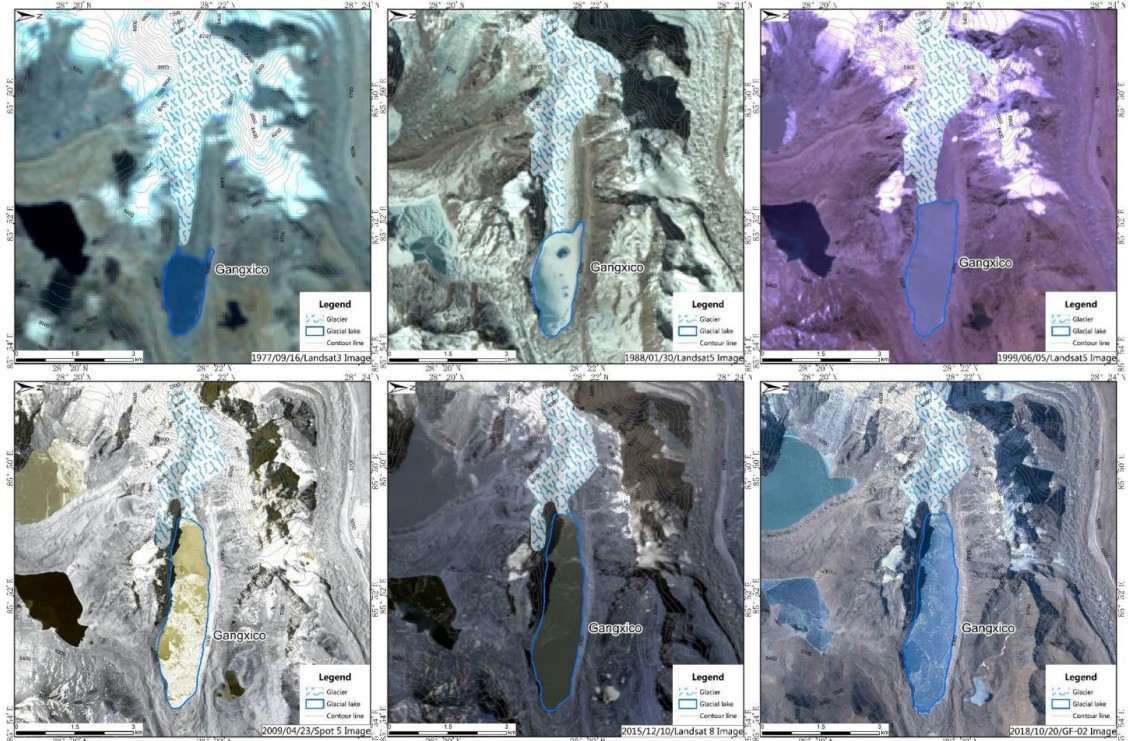

**Fig. 9 Comparison of area change between Ganxico Lake and its connected glacier**





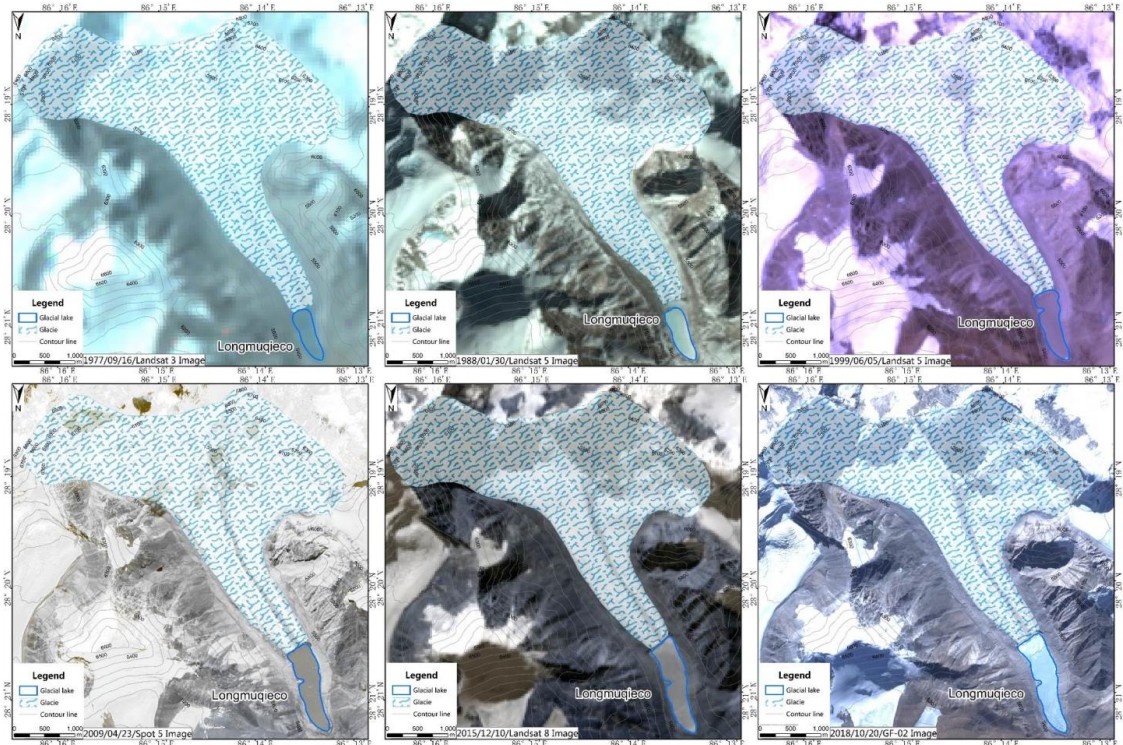

**Fig. 10 Comparison of area change between Longmuqieco Lake and its connected glacier**





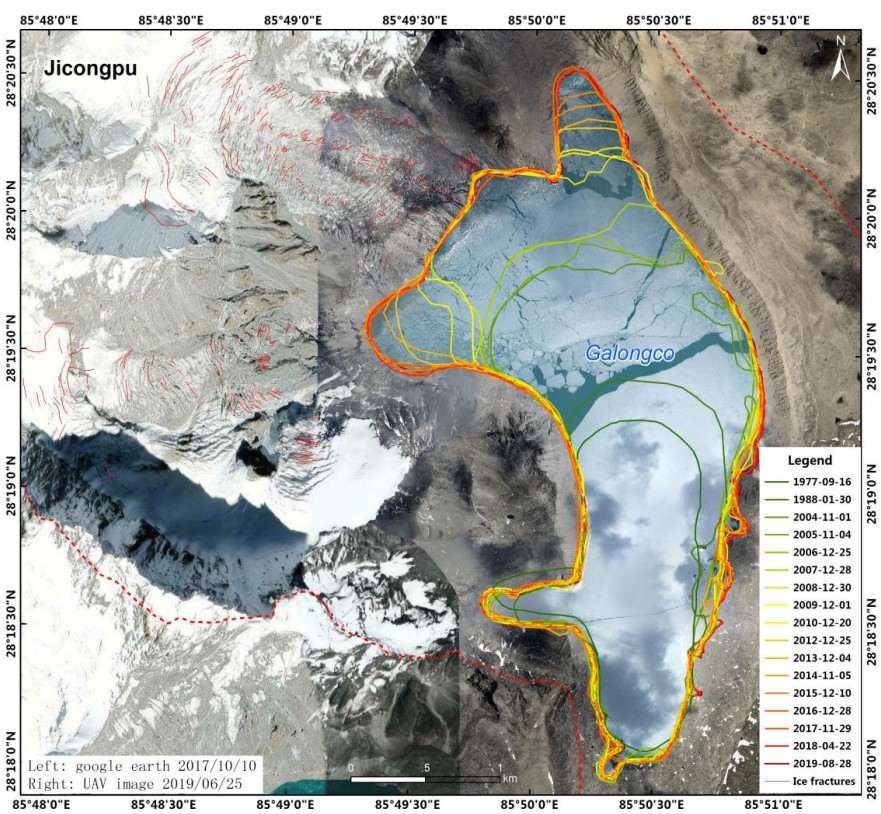

**Fig. 11 Variation in the area of Galongco Lake (1977-2019) (The left image is from © Google Earth and the right image about the Galongco Lake is from UAV image)**



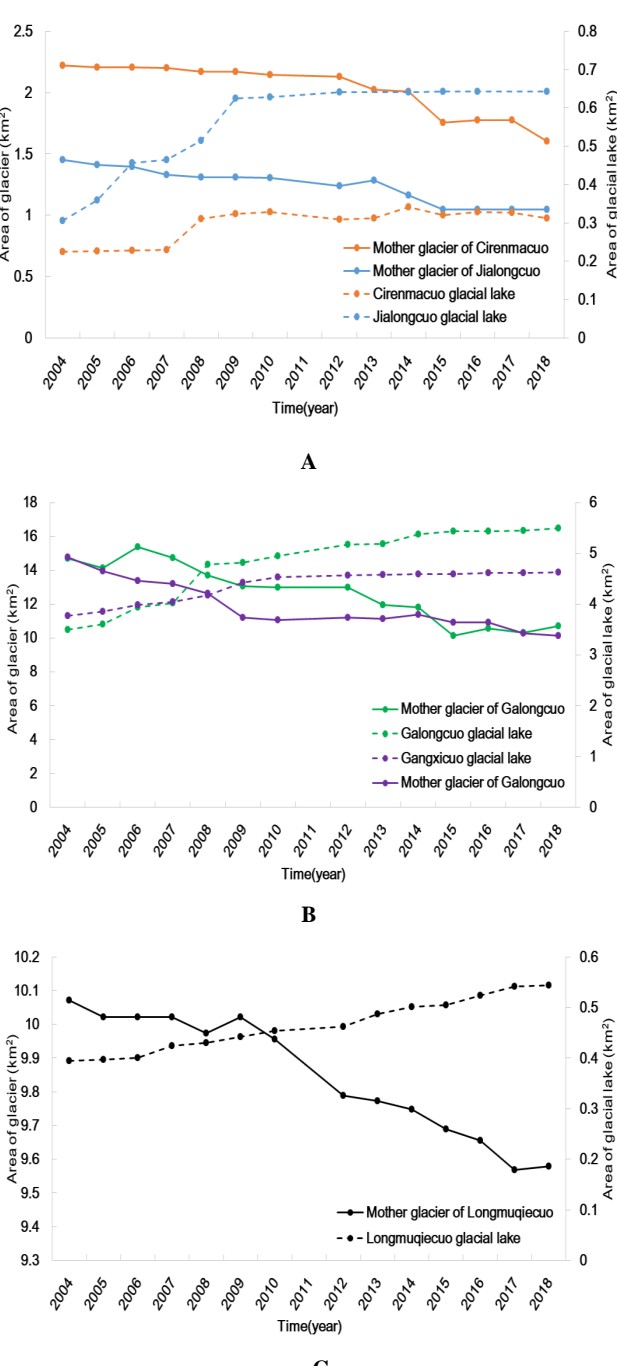

**Fig. 12 Retreat of 5 typical glaciers, growth of 5 typical glacial lakes and rates of change in**

**the Poiqu River Basin**



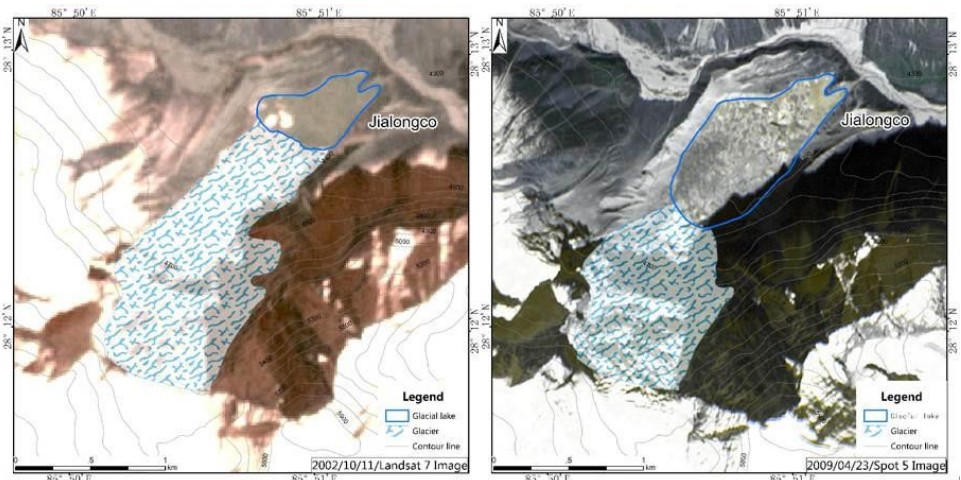

**Fig. 13 Rapid rise in Jialongco Lake due to glacial loss**

**(2002-2009)**



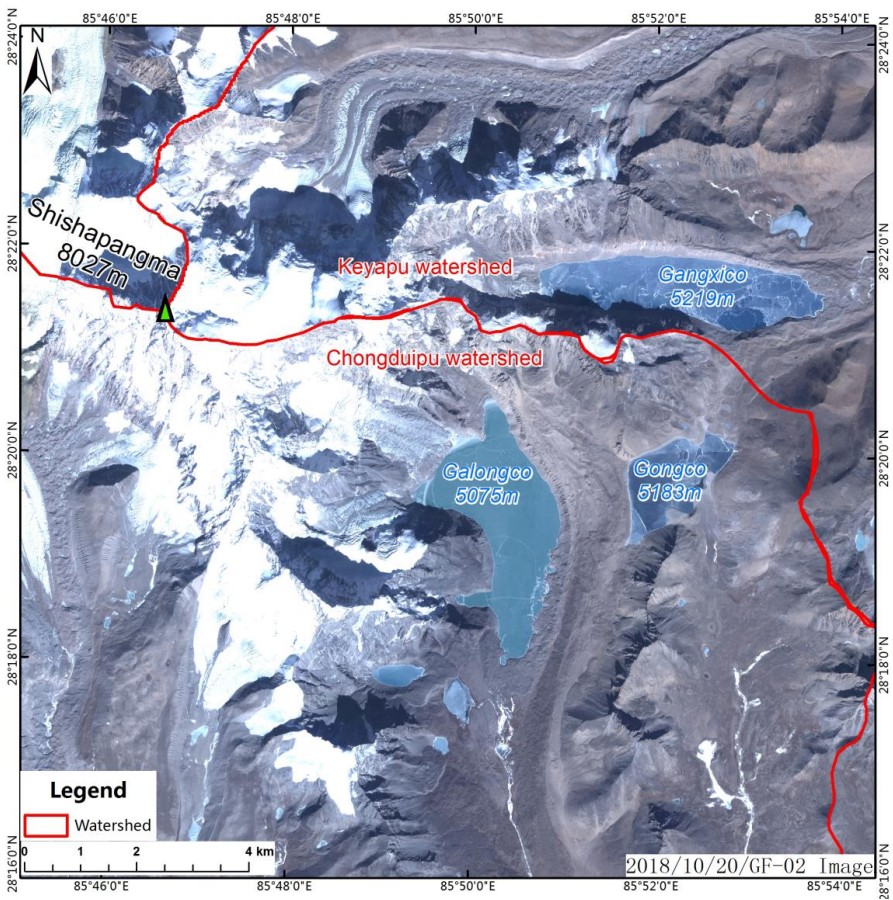

**Fig. 14 Hydraulically connected glacial lakes (Galongco, Gangco, and Gangxico)**





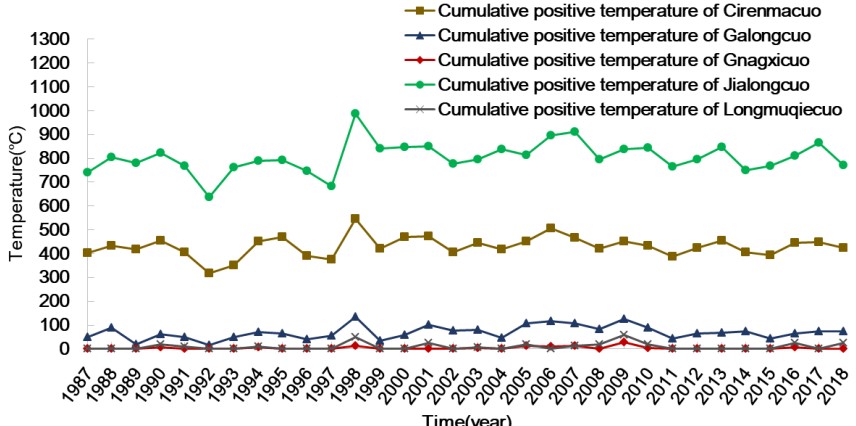

**Fig. 15A. Interpolated cumulative temperature**

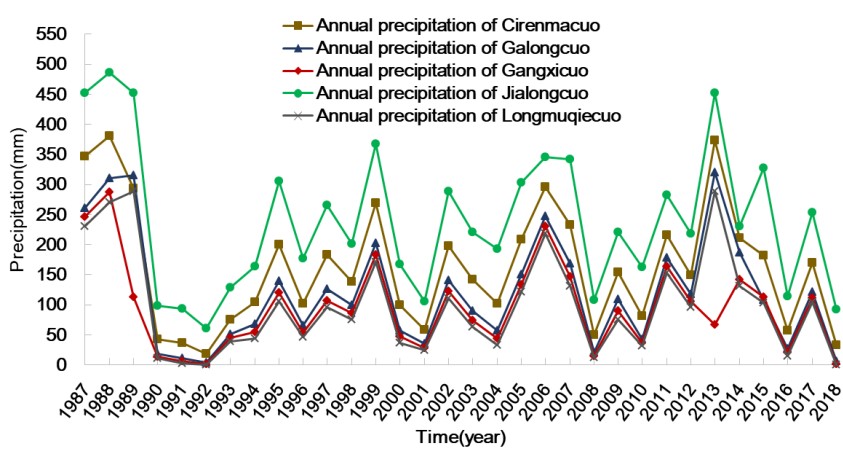

**Fig. 15B. Interpolated annual precipitation**

**Fig. 15 Interpolated temperatures and precipitation for the glacial lakes**

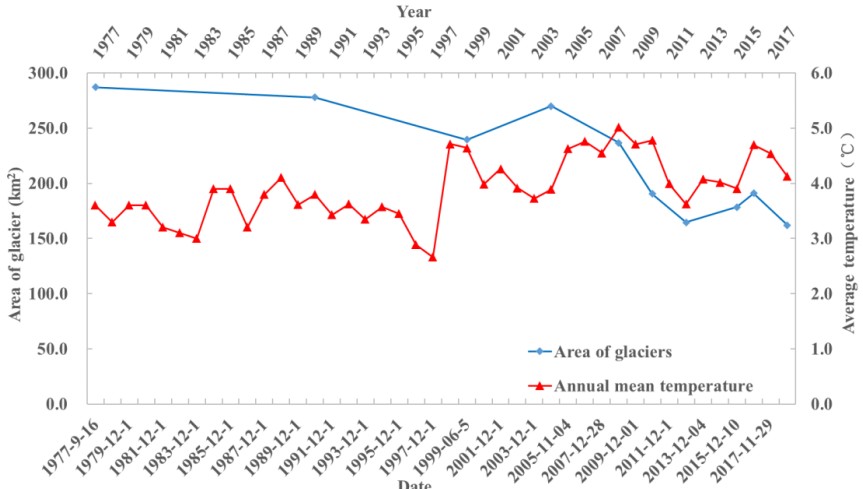

**Fig. 16A Changes in the area of glaciers vs. temperature**

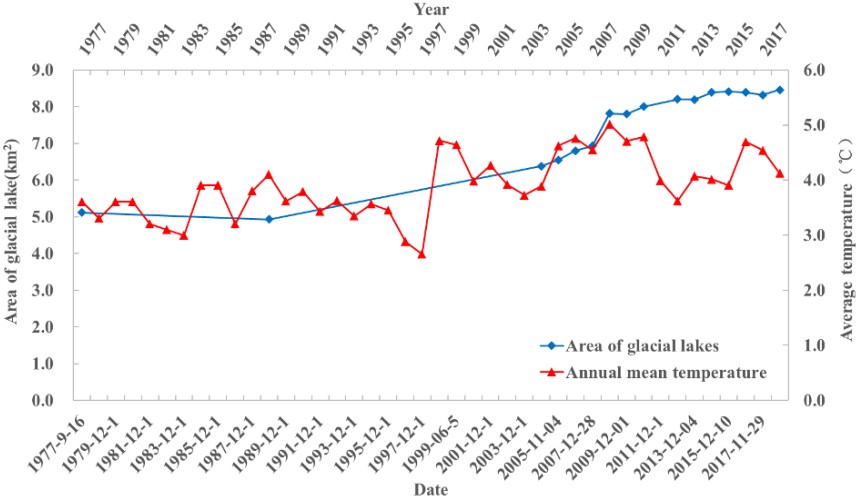

**Fig. 16B Changes in the area of glacial lakes vs. temperature**

**Fig. 16 Changes in the area of glaciers and glacial lakes vs. temperature in Poiqu**



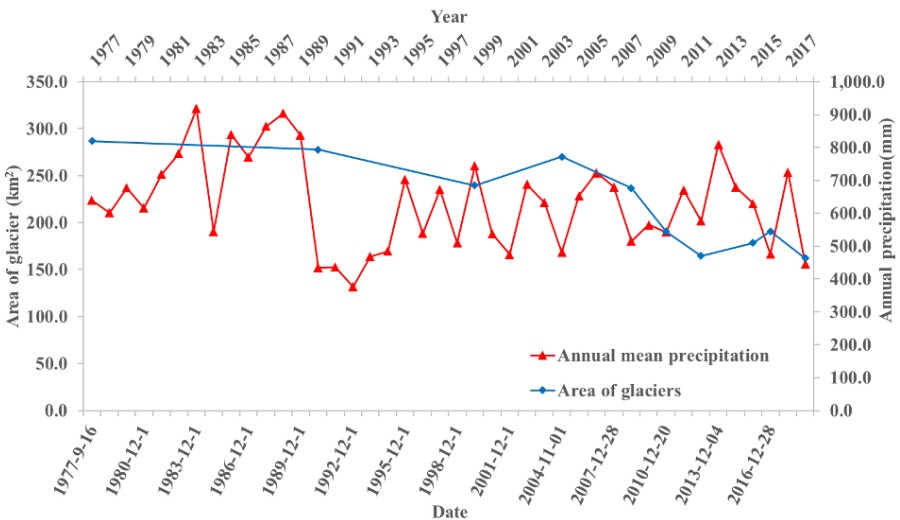

**Fig. 17A Changes in the area of glaciers vs. precipitation**

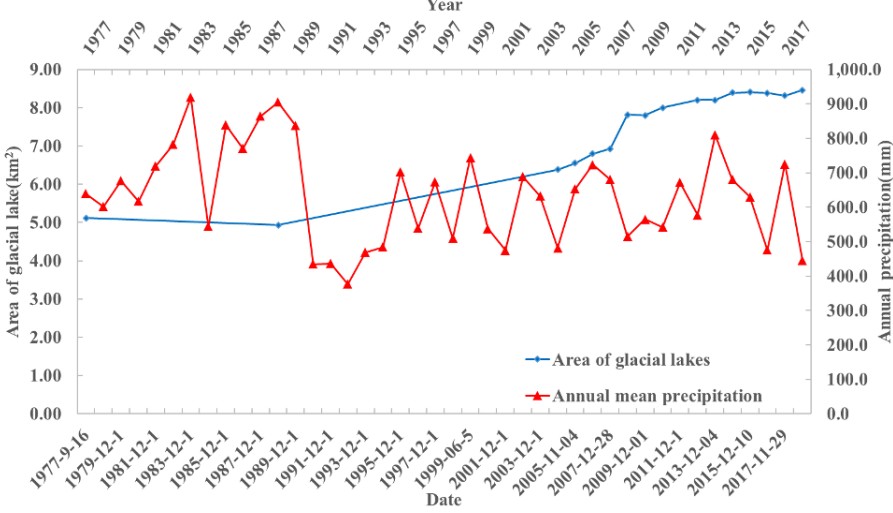

**Fig. 17B Changes in the area of glacial lakes vs. precipitation**

**Fig. 17 Changes in the area of glaciers and glacial lakes vs. precipitation in Poiqu**

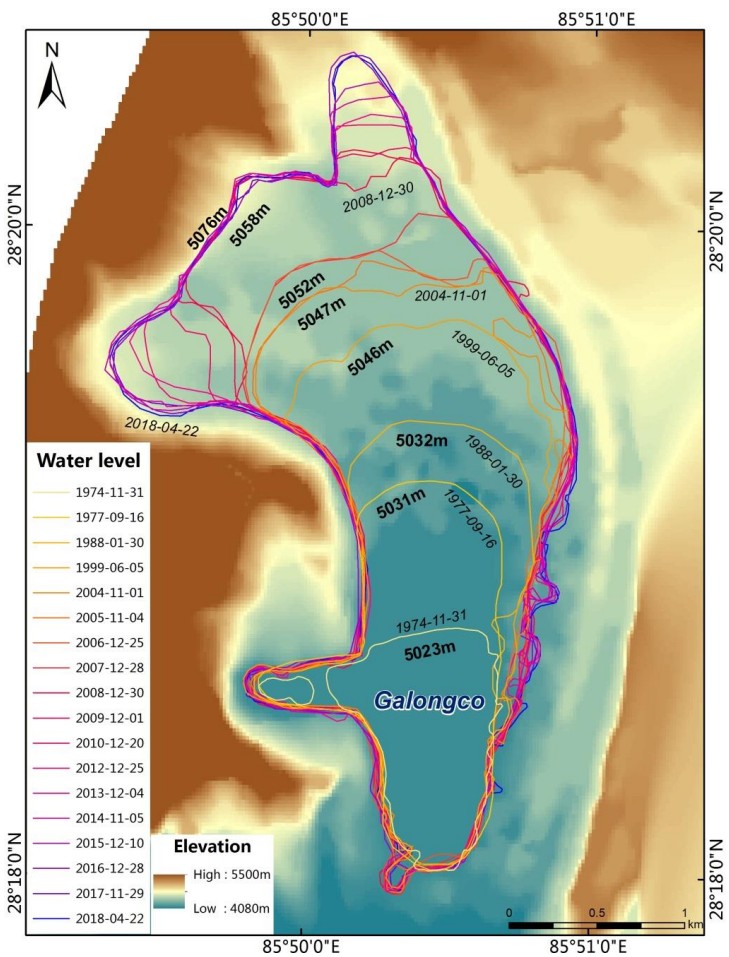

**Fig. 18 Terrain reconstruction of GB Lake below the water level**





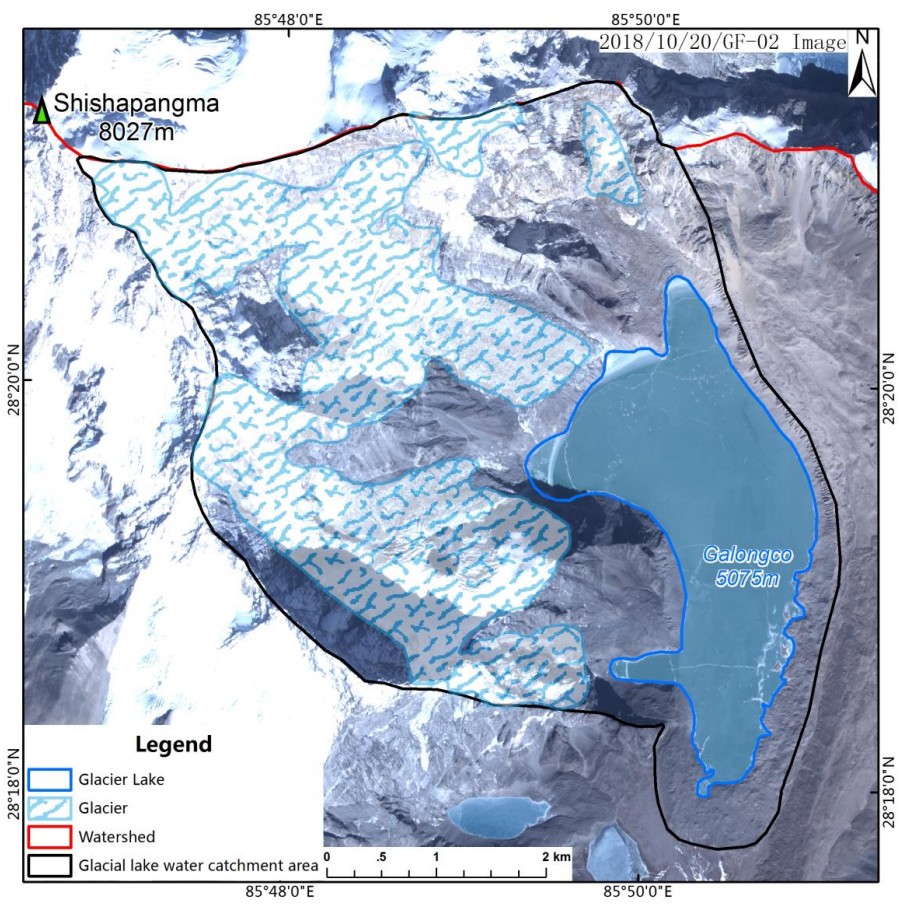

**Fig. 19 Galongco Lake and the connected glaciers in 2006 (The image is from GF-02 image)**

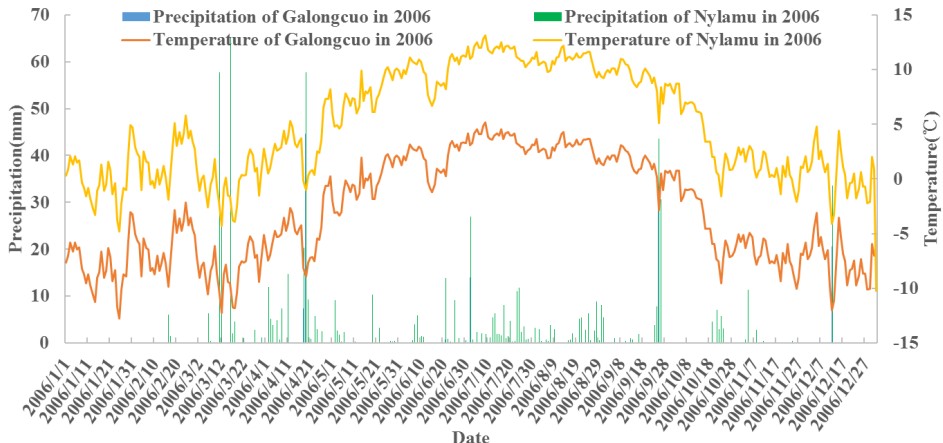



**Fig. 20 Temperature and precipitation of Galongco Lake in 2006**

**Table 1 Data sources and features for interpretation of glaciers and glacial lakes**

| Satellites | Spot number | Date | Sensors | Spectrum features | Spatial Resolution(m) |
|---|---|---|---|---|---|
| Landsat 3 | LM21510401977259AAA01 | 1977-09-16 | MSS | 4 bands, from visible to near infrared | Multi-Spectral (60m) |
| Landsat 5 | LT51410401988030BKT00 | 1988-01-30 | TM | 7 bands, from visible to near infrared | Multi-Spectral (30m) |
| Landsat 7 | LE71410402004306PFS00 | 2004-11-01 | ETM | 7 bands, from visible to intermediate infrared Micron Panchromatic | Multi-Spectral (30m) Panchromatic (15m) |
| Landsat 7 | LE71410402005308PFS00 | 2005-11-04 | | | |
| Landsat 7 | LE71410402006359SGS00 | 2006-12-25 | | | |
| Landsat 7 | LE71410402007362SGS00 | 2007-12-28 | | | |
| Landsat 7 | LE71410402008365SGS00 | 2008-12-30 | | | |
| Landsat 7 | LE71410402009335SGS00 | 2009-12-01 | | | |
| Landsat 7 | LE71410402010354PFS00 | 2010-12-20 | | | |
| Landsat 7 | LE71410402012360PFS00 | 2012-12-25 | | | |
| ASTER GDEM | ASTGTM_N27E085/ N27E086/ N28E085/ N28E086 | 2009 | ASTER | 14 bands, 3 visible/ near infrared, 6 short-wave infrared, 3 thermal infrared band | visible/ near infrared (15m), short-wave infrared (30m), 3 thermal infrared (90m) |
| SPOT-5 | S5G1B201004230520204YZYZMX S5G1J201004230520206YZYZMX S5G1A201004230520201YZYZMX | 2010-04-23 | HRG$_s$ | 5 bands, 1 Panchromatic, 1 short-wave infrared, 3 Multi-Spectral | Panchromatic (2.5m), Multi-Spectral (10m), short-wave infrared (20m) |
| Landsat 8 | LC81410402013338LGN00 | 2013-12-04 | OLI | 7 bands, from visible to intermediate infrared, 2 thermal infrared, Panchromatic, Cirrus | Multi-Spectral (30m), Multi-Spectral(30m), Cirrus, Thermal infrared (100m), Panchromatic (15m) |
| Landsat 8 | LC81410402014309LGN02 | 2014-11-05 | | | |
| Landsat 8 | LC81410402015344LGN00 | 2015-12-10 | | | |
| Landsat 8 | LC81410402016363LGN00 | 2016-12-28 | | | |
| Landsat 8 | LC81410402017333LGN00 | 2017-11-29 | | | |
| Landsat 8 | LC81410402018112LGN00 | 2018-04-22 | | | |
| GF-2 | L1A0003537778/GF2_PMS2_3537778 | 2018-10-20 | MSS | 4 bands, from visible to near infrared, Panchromatic | Panchromatic (1m), Multi-Spectral (4m) |
| GF-2 | L1A0002952275/GF2_PMS2_2952275 | 2018-01-22 | | | |
| GF-2 | L1A0002952269/GF2_PMS2_2952269 | 2018-01-22 | | | |
| GF-2 | L1A0002951335/GF2_PMS2_2951335 | 2018-01-22 | | | |
| GF-2 | L1A0002951338/GF2_PMS2_2951338 | 2018-01-22 | | | |
| UAV | SONY Alpha 7 III (4) | 2019-05-30 | Visible light | | |
| ZY-3 | 52252940901110452051A 52252931103010521012U | 2009/01/11 2011/03/01 | TLC | 4 bands, from visible to near infrared Foresight, back sight, Panchromatic | Foresight, Backsight (3.5m); Orthophoto (2.1m), Multi-Spectral (3.5m) |



**Table 2 Interpretation signs for glaciers and glacial lakes(Six pictures are from © Google Earth**

**images and two pictures are from GF-2 images. They are signed in the lower right corner)**

| Types | Moraine lake | Ice surface lake | Cirque lake | Glacial erosion lake |
|---|---|---|---|---|
| Images | | | | |
| Signs | Formed by pool eroded by glacier. | Occurring in the melted area of glacier covered by surface moraines. | Forming in the cirque, with steep rocky walls. | Having gentle bank with residual boulders. |
| Types | Valley glacier | Cirque glacier | Hanging glacier | Moraine |
| Images | | | | |
| Signs | Located in low-lying gorges, with white or blue tone and irregular plane shapes, having thick front tongue. | In chair-shaped hollow in the source slope, in moderate size between 1 - 10km$^2$. | Hanging isolatedly on slope near the peak, thin and small. | Moraine is glacially formed accumulation of unconsolidated glacial debris (regolith and rock). |

**Table 3 Types of glacial lakes in the Poiqu River Basin**

| Types | Moraine lake | Glacier-eroded lake | Glacier-surface lake | Cirque lake |
|---|---|---|---|---|
| Numbers | 19 | 84 | 20 | 24 |
| Percentage | 13% | 57% | 14% | 16% |
| Area (km$^2$) | 15.2 | 1.7 | 0.40 | 2.0 |





**Table 4 Typical glacial lakes in tributaries of the Poiqu River Basin**

| Tributaries | Lakes | Types | Longitude E (°) | Latitude N(°) | Area (km²) | Altitude (m) | Distance to mother glacier (km) |
|---|---|---|---|---|---|---|---|
| Chongduipu | Galongco | EM | 85.8382 | 28.3222 | 5.5 | 5076 | 0.18 |
| | gongco | ER | 85.8693 | 28.3293 | 2.13 | 5183 | without glacier |
| | Jialongco | EM | 85.8475 | 28.211 | 0.6 | 4382 | 0 |
| | Dareco | ER | 85.9229 | 28.1816 | 0.48 | 4372 | without glacier |
| | Anonymous 11 | EM | 85.8197 | 28.2975 | 0.29 | 5092 | 0.1 |
| | Anonymous P | V | 85.8307 | 28.2936 | 0.28 | 5009 | without glacier |
| | Suannaihu | EM | 85.9059 | 28.1507 | 0.13 | 4507 | 1.5 |
| | Anonymous Q | EM | 85.9198 | 28.1386 | 0.1 | 4882 | 0 |
| Dajilingpu | Paquco | EM | 86.1575 | 28.3035 | 0.6 | 5306 | 0 |
| | Tananco | EM | 86.151 | 28.2953 | 0.17 | 5337 | 0 |
| Duogapu | Tuzhuoco | EM | 86.1032 | 28.2532 | 0.12 | 5208 | 0.1 |
| Gangpu | Gangpuco | EM | 86.1586 | 28.321 | 0.22 | 5539 | 0.5 |
| Karupu | Longjueco | V | 85.9155 | 28.2595 | 0.25 | 5103 | without glacier |
| Keyapu | Gangxico | EM | 85.8708 | 28.36 | 4.6 | 5219 | 0 |
| | Anonymous 8 | EM | 85.9488 | 28.3141 | 0.31 | 5223 | 0.2 |
| | Yingreco | EM | 85.8907 | 28.3712 | 0.27 | 5225 | without glacier |
| | Mabiya | EM | 85.9079 | 28.3234 | 0.15 | 5419 | 0.4 |
| | Anonymous X | EM | 85.9304 | 28.3213 | 0.11 | 5319 | 0 |
| | Chamaqudan | EM | 86.1921 | 28.3352 | 0.54 | 5420 | 0.3 |
| Rujiapu | Longmuqieco | EM | 86.2259 | 28.3468 | 0.52 | 5342 | 0 |
| | Dareco | EM | 86.1314 | 28.2941 | 0.21 | 5233 | 0.2 |
| Zhangzangbu | Cirenmaco | EM | 86.0664 | 28.067 | 0.33 | 4639 | 0.29 |

Note: EM-- End moraine-dammed lake; ER--Glacial erosion lake; V—Glacial valley lake.

**Table 5 Parameters of the glacial lake tributaries**

| Tributaries | Area (km²) | Glacier number | Average slope | Elevation difference (m) | Moraine (km²) | Glacier area (km²) | Lake area (km²) |
|---|---|---|---|---|---|---|---|
| Chongduipu | 372.77 | 55 | 23.7° | 4277 | 64.1 | 68.66 | 10.44 |
| Zhangzangbu | 49.92 | 14 | 29.3° | 2941 | 9.2 | 8.28 | 0.42 |
| Rujiapu | 354.89 | 13 | 21.9° | 2636 | 7.1 | 27.33 | 1.63 |
| Keyapu | 163.96 | 25 | 18.7° | 3807 | 29.2 | 27.5 | 5.87 |





**Table 6 Basic parameters for major glacial lakes in the Poiqu River Basin**

| Major lakes | Tributaries | Water supply area (km²) | Connected glacier (km²) | Distance to glacier (km) | Water level altitude (m) |
|---|---|---|---|---|---|
| Galongco | Chongduipu | 29.61 | 10.71 | 0.18 | 5076 |
| Jialongco | Chongduipu | 5.61 | 0.88 | 0 | 4382 |
| Longmuqieco | Rujiapu | 19.30 | 9.58 | 0 | 5342 |
| Cirenmaco | Zhangzangbu | 5.10 | 1.61 | 0.29 | 4639 |
| Gangxico | Keyapu | 15.91 | 3.38 | 0 | 5219 |

**Table 7 Area variations and annual speeds of glaciers and glacial lakes in Poiqu river Basin since 1977**

| Year | Area (km²) | | Annual speeds of change from 1977 to 2016 (km²/a) | | Annual speeds of change from 2004 to 2016 (km²/a) | |
|---|---|---|---|---|---|---|
| | Glacier | Glacial lake | Glacier | Glacial lake | Glacier | Glacial lake |
| 1977 | 287.00 | 5.12 | | | | |
| 2004 | 270.10 | 6.38 | | | | |
| 2008 | 236.70 | 7.82 | -2.46 | +0.08 | -6.60 | +0.17 |
| 2010 | 190.4 | 8.00 | | | | |
| 2015 | 178.4 | 8.41 | | | | |
| 2016 | 190.9 | 8.38 | | | | |





**Table 8 Area variations in 5 typical glacial lakes and their glaciers since 1977**

| Date | Cirenmaco | | Galongco | | Jialongco | | Gangxico | | Longmuqieco | |
|---|---|---|---|---|---|---|---|---|---|---|
| | Area of Glacier (km²) | Area of Glacial Lake (km²) | Area of Glacier (km²) | Area of Glacial Lake (km²) | Area of Glacier (km²) | Area of Glacial Lake (km²) | Area of Glacier (km²) | Area of Glacial Lake (km²) | Area of Glacier (km²) | Area of Glacial lake (km²) |
| 1977/9/16 | 2.58 | 0.57 | 17.15 | 1.66 | 1.81 | 0.10 | 6.13 | 1.82 | 10.6 | 0.22 |
| 1988/1/30 | 2.24 | 0.12 | 14.91 | 2.07 | 1.47 | 0.15 | 5.49 | 2.53 | 10.25 | 0.24 |
| 1999/6/5 | 2.04 | 0.19 | 14.78 | 2.98 | 1.48 | 0.20 | 5.12 | 3.31 | 10.19 | 0.35 |
| 2002/12/1 | | | | | | 0.24 | | | | |
| 2002/12/1 | | | | | | 0.30 | | | | |
| 2004/11/01 | 2.22 | 0.23 | 14.72 | 3.50 | 1.45 | 0.31 | 4.92 | 3.77 | 10.07 | 0.39 |
| 2005/11/04 | 2.21 | 0.23 | 14.12 | 3.60 | 1.41 | 0.36 | 4.65 | 3.86 | 10.02 | 0.40 |
| 2006/12/25 | 2.21 | 0.23 | 45.37 | 3.93 | 1.39 | 0.46 | 4.46 | 3.99 | 10.02 | 0.40 |
| 2007/12/28 | 2.20 | 0.31 | 14.73 | 4.01 | 1.33 | 0.46 | 4.40 | 4.04 | 10.02 | 0.42 |
| 2008/12/30 | 2.17 | 0.32 | 13.71 | 4.78 | 1.31 | 0.51 | 4.22 | 4.17 | 9.97 | 0.43 |
| 2009/12/1 | 2.17 | 0.32 | 13.05 | 4.81 | 1.31 | 0.63 | 3.73 | 4.41 | 10.02 | 0.44 |
| 2010/12/20 | 2.14 | 0.33 | 12.98 | 4.95 | 1.30 | 0.63 | 3.68 | 4.54 | 9.96 | 0.45 |
| 2012/12/25 | 2.13 | 0.31 | 12.99 | 5.17 | 1.24 | 0.64 | 3.74 | 4.56 | 9.79 | 0.46 |
| 2013/12/04 | 2.02 | 0.31 | 11.96 | 5.18 | 1.28 | | 3.71 | 4.58 | 9.77 | 0.49 |
| 2014/11/05 | 2.01 | 0.34 | 11.81 | 5.38 | 1.16 | 0.64 | 3.79 | 4.59 | 9.74 | 0.50 |
| 2015/12/10 | 1.75 | 0.32 | 10.15 | 5.43 | 1.05 | 0.64 | 3.64 | 4.60 | 9.69 | 0.50 |
| 2016/12/28 | 1.77 | 0.33 | 10.54 | 5.44 | 1.05 | 0.64 | 3.64 | 4.61 | 9.66 | 0.52 |
| 2017/11/29 | 1.78 | 0.33 | 10.31 | 5.45 | 1.05 | | 3.42 | 4.62 | 9.67 | 0.54 |
| 2018/4/22 | 1.61 | 0.31 | 10.71 | 5.50 | 1.05 | 0.64 | 3.38 | 4.63 | 9.58 | 0.54 |





**Table 9 Annual rates of change in 5 typical glacial lakes and their glaciers**

| Glacial lake | Annual speed of change 1997-2018 (km²) | | Annual speed of change 2004-2018 (km²) | |
|---|---|---|---|---|
| | Area of Glacier | Area of Glacial lake | Area of Glacier | Area of Glacial lake |
| Cirenmacuo | -0.018 | -0.006 | -0.044 | +0.006 |
| Galongco | -0.167 | +0.093 | -0.286 | +0.143 |
| Jialongco | -0.018 | +0.013 | -0.029 | +0.024 |
| Gangxico | -0.066 | 0.068 | -0.110 | 0.061 |
| Longmuqieco | -0.015 | 0.008 | -0.035 | 0.011 |

**Table 10 Parameters for the water balance calculation of glacial lakes**

| Glaciers | Runoff coefficient | $R_c$ for snow cover | $R_c$ for glacier | DDF (snow) | DDF (glacier) | Drainage area to lake (km²) |
|---|---|---|---|---|---|---|
| Cirenmaco | 0.60 | 0.60 | 0.53 | 8.30 | 12.60 | 9.77 |
| Galongco | 0.56 | 0.56 | 0.50 | 8.30 | 12.60 | 22.33 |
| Gangxico | 0.54 | 0.54 | 0.47 | 8.30 | 12.60 | 19.1 |
| Jialongco | 0.61 | 0.61 | 0.56 | 6.70 | 9.60 | 5.76 |
| Longmeqieco | 1.00 | 1.00 | 1.00 | 7.40 | 11.60 | 19.47 |





**Table 11 Water balance for Galongco Lake between 1988 and 2018**

| Year | Glacier area (km²) | Rainfall (mm) | Runoff (10⁴m³) | $T_c$ (°C) | $T_{cG}$ (°C) | $M_G$ (mm) | $W_G$ (10⁴m) | Snowfall (mm) | $T_{cS}$ (°C) | $W_{snow}$ (10⁴m³) | $W_{total}$ (10⁴m³) | Infiltration (10⁴m³) |
|---|---|---|---|---|---|---|---|---|---|---|---|---|
| 1987 | 21.3 | 16.5 | 18.4 | 210.4 | 12.4 | 112.4 | 119.9 | 253.5 | 198.0 | 226.4 | 364.7 | 404.5 |
| 1988 | 21.0 | 1.5 | 1.7 | 233.4 | 0.0 | 0.0 | 0.0 | 309.0 | 241.4 | 276.0 | 277.7 | 414.4 |
| 1989 | 20.6 | 7.4 | 8.3 | 204.0 | 0.0 | 0.0 | 0.0 | 308.0 | 240.6 | 275.1 | 283.4 | 424.2 |
| 1990 | 20.2 | 0.0 | 0.0 | 252.8 | 101.0 | 919.1 | 930.5 | 194.3 | 151.8 | 16.3 | 946.8 | 434.1 |
| 1991 | 19.9 | 5.4 | 6.0 | 214.5 | 85.8 | 780.3 | 775.9 | 164.8 | 128.8 | 5.4 | 787.4 | 444.0 |
| 1992 | 19.5 | 2.3 | 2.6 | 142.1 | 67.4 | 613.5 | 598.9 | 95.6 | 74.7 | 0.0 | 601.4 | 453.8 |
| 1993 | 19.2 | 0.0 | 0.0 | 190.8 | 76.5 | 696.2 | 667.0 | 146.3 | 114.3 | 46.1 | 713.1 | 463.7 |
| 1994 | 18.8 | 4.1 | 4.6 | 257.9 | 107.2 | 975.5 | 917.0 | 192.9 | 150.7 | 57.0 | 978.6 | 473.5 |
| 1995 | 18.4 | 23.3 | 26.0 | 253.5 | 118.4 | 1077.6 | 993.6 | 172.9 | 135.1 | 103.7 | 1123.3 | 483.4 |
| 1996 | 18.1 | 7.1 | 7.9 | 207.2 | 78.5 | 714.6 | 646.0 | 164.7 | 128.7 | 52.2 | 706.0 | 493.2 |
| 1997 | 17.7 | 19.2 | 21.4 | 199.3 | 84.6 | 770.0 | 682.1 | 146.8 | 114.7 | 95.0 | 798.6 | 503.1 |
| 1998 | 17.4 | 13.0 | 14.5 | 326.1 | 121.9 | 1109.1 | 962.5 | 261.4 | 204.2 | 77.6 | 1054.6 | 512.9 |
| 1999 | 17.0 | 27.3 | 30.5 | 218.4 | 81.1 | 737.6 | 626.7 | 175.8 | 137.3 | 157.0 | 814.3 | 522.8 |
| 2000 | 16.6 | 3.0 | 3.3 | 257.0 | 88.5 | 805.2 | 669.6 | 215.7 | 168.5 | 48.9 | 721.8 | 532.6 |
| 2001 | 16.3 | 8.2 | 9.2 | 256.2 | 69.4 | 631.6 | 513.8 | 239.1 | 186.8 | 24.4 | 547.4 | 542.5 |
| 2002 | 15.9 | 0.0 | 0.0 | 229.1 | 93.3 | 849.2 | 675.5 | 173.8 | 135.8 | 125.5 | 801.0 | 552.3 |
| 2003 | 15.5 | 22.9 | 25.6 | 260.6 | 129.7 | 1180.6 | 917.8 | 167.5 | 130.9 | 49.8 | 993.1 | 562.2 |
| 2004 | 14.7 | 21.0 | 23.4 | 235.8 | 164.3 | 1495.3 | 1100.2 | 91.5 | 71.5 | 41.7 | 1165.4 | 572.0 |
| 2005 | 14.1 | 24.7 | 27.6 | 266.2 | 147.0 | 1337.5 | 944.5 | 152.6 | 119.2 | 112.3 | 1084.4 | 581.9 |
| 2006 | 15.4 | 14.0 | 15.6 | 282.3 | 116.0 | 1055.3 | 810.8 | 212.9 | 166.3 | 190.2 | 1016.6 | 591.8 |
| 2007 | 14.7 | 7.5 | 8.4 | 285.7 | 143.9 | 1309.5 | 964.7 | 181.5 | 141.8 | 162.1 | 1135.2 | 601.6 |
| 2008 | 13.7 | 8.6 | 9.6 | 249.1 | 147.9 | 1346.1 | 923.3 | 129.5 | 101.2 | 11.3 | 944.2 | 611.4 |
| 2009 | 13.1 | 3.6 | 4.0 | 261.5 | 142.3 | 1294.8 | 845.0 | 152.6 | 119.2 | 94.0 | 942.9 | 621.3 |
| 2010 | 13.0 | 2.9 | 3.2 | 258.4 | 165.7 | 1508.3 | 978.6 | 118.6 | 92.7 | 35.5 | 1017.3 | 631.1 |
| 2011 | 12.7 | 41.2 | 46.0 | 205.0 | 128.3 | 1167.4 | 738.7 | 98.2 | 76.7 | 122.5 | 907.3 | 641.0 |
| 2012 | 13.0 | 14.5 | 16.2 | 240.3 | 152.2 | 1384.8 | 899.7 | 112.8 | 88.1 | 91.8 | 1007.7 | 650.8 |
| 2013 | 12.0 | 8.9 | 9.9 | 258.1 | 164.6 | 1497.7 | 895.5 | 119.7 | 93.5 | 278.5 | 1184.0 | 660.7 |
| 2014 | 11.8 | 35.0 | 39.1 | 235.9 | 160.8 | 1463.5 | 864.3 | 96.1 | 75.1 | 67.3 | 970.8 | 670.5 |
| 2015 | 10.2 | 0.6 | 0.7 | 211.0 | 129.3 | 1176.5 | 597.2 | 104.6 | 81.7 | 157.6 | 755.5 | 680.4 |
| 2016 | 10.5 | 10.4 | 11.6 | 248.0 | 168.0 | 1528.3 | 805.9 | 102.4 | 80.0 | 14.8 | 832.4 | 690.3 |
| 2017 | 10.3 | 12.0 | 13.4 | 221.5 | 147.8 | 1345.4 | 693.5 | 94.3 | 73.7 | 98.1 | 805.0 | 700.1 |
| 2018 | 10.7 | 4.2 | 4.7 | 260.6 | 181.3 | 1649.9 | 883.4 | 101.5 | 79.3 | 1.9 | 890.0 | 710.0 |





**Table 12 Comparison between the calculated water quantity and the observed quantity**

| | Cirenmaco | | | Galongco | | | Gangxico | | | Jialongco | | | Longmuqieco | | |
|------|------|------|------|------|------|------|------|------|------|------|------|------|------|------|------|
| Year | MV | TV | ER(%) | MV | TV | ER(%) | MV | TV | ER(%) | MV | TV | ER(%) | MV | TV | ER(%) |
| 1988 | 0.180 | 0.224 | 24.44 | 0.369 | 0.367 | -0.54 | 0.355 | 0.363 | 1.75 | 0.087 | 0.105 | 20.69 | 0.247 | 0.252 | 2.50 |
| 1999 | 0.412 | 0.346 | -16.02 | 0.526 | 0.551 | 4.94 | 0.523 | 0.565 | 8.60 | 0.121 | 0.353 | 191.74 | 0.464 | 0.455 | -1.80 |
| 2000 | | 0.425 | | | 0.568 | | | 0.571 | | 0.157 | 0.392 | 149.04 | | 0.457 | |
| 2001 | | 0.474 | | | 0.579 | | | 0.586 | | | 0.427 | | | 0.459 | |
| 2002 | | 0.498 | | | 0.58 | | | 0.633 | | 0.187 | 0.459 | 145.45 | | 0.516 | |
| 2003 | | 0.492 | | | 0.594 | | | 0.648 | | 0.273 | 0.481 | 76.19 | | 0.535 | |
| 2004 | 0.571 | 0.519 | -9.11 | 0.598 | 0.619 | 3.68 | 0.655 | 0.673 | 2.43 | 0.281 | 0.51 | 81.49 | 0.576 | 0.540 | -6.00 |
| 2005 | 0.579 | 0.551 | -4.84 | 0.538 | 0.654 | 21.56 | 0.684 | 0.688 | 0.57 | 0.37 | 0.543 | 46.76 | 0.582 | 0.545 | -6.17 |
| 2006 | 0.581 | 0.608 | 4.48 | 0.566 | 0.683 | 20.67 | 0.729 | 0.714 | -2.00 | 0.557 | 0.569 | 1.97 | 0.591 | 0.555 | -6.00 |
| 2007 | 0.590 | 0.694 | 17.63 | 0.63 | 0.708 | 12.38 | 0.748 | 0.739 | -1.29 | 0.574 | 0.607 | 5.57 | 0.653 | 0.597 | -7.86 |
| 2008 | 0.991 | 0.763 | -23.11 | 0.72 | 0.739 | 2.64 | 0.797 | 0.763 | -4.25 | 0.684 | 0.645 | -5.70 | 0.670 | 0.609 | -8.71 |
| 2009 | 1.062 | 0.766 | -27.87 | 0.713 | 0.758 | 6.31 | 0.901 | 0.797 | -11.56 | 0.954 | 0.675 | -29.25 | 0.701 | 0.656 | -6.43 |
| 2010 | 1.090 | 0.807 | -25.96 | 0.687 | 0.777 | 13.10 | 0.954 | 0.847 | -10.80 | 0.962 | 0.706 | -26.61 | 0.734 | 0.761 | 3.86 |
| 2011 | | 0.826 | | | 0.8 | | | 0.888 | | | 0.74 | | | 0.785 | |
| 2012 | 0.986 | 0.854 | -13.39 | 0.691 | 0.815 | 17.95 | 0.967 | 0.898 | -6.90 | 0.997 | 0.768 | -22.87 | 0.757 | 0.790 | 4.13 |
| 2013 | 1.002 | 0.882 | -11.98 | 0.689 | 0.836 | 21.34 | 0.973 | 0.910 | -6.30 | | 0.795 | | 0.827 | 0.796 | -3.88 |
| 2014 | 1.169 | 0.957 | -18.14 | 0.761 | 0.866 | 13.80 | 0.980 | 0.928 | -5.20 | 0.997 | 0.82 | -17.75 | 0.869 | 0.805 | -7.00 |
| 2015 | 1.042 | 0.986 | -5.37 | 0.803 | 0.884 | 10.09 | 0.983 | 0.944 | -4.00 | 0.999 | 0.84 | -15.92 | 0.880 | 0.820 | -6.67 |
| 2016 | 1.094 | 0.987 | -9.78 | 0.915 | 0.888 | -2.95 | 0.991 | 0.952 | -3.90 | 1 | 0.854 | -14.60 | 0.940 | 0.826 | -12.67 |
| 2017 | 1.081 | 1.006 | -6.94 | 0.945 | 0.896 | -5.08 | 0.993 | 0.984 | -0.90 | | 0.882 | | 0.993 | 0.893 | -10.00 |
| 2018 | 1.000 | 1.055 | 5.5 | 1 | 0.903 | -9.70 | 1.000 | 1.000 | 0.00 | 1 | 0.915 | -8.50 | 1.000 | 0.906 | -9.40 |

Illustration: MV- Measured Value, TV-Theoretical Value, ER- Error Rate



**Table 13 Fractions of various water supplies to the lakes**

| Glacial lakes | Elevation (m) | Supply | | | Loss |
| --- | --- | --- | --- | --- | --- |
| | | Glacier (%) | Snow (%) | Rainfall (%) | Seepage Flow (%) |
| Cirenmaco | 4642 | 78.3 | 17.3 | 4.4 | 95.4 |
| Galongco | 5075 | 88.0 | 11.1 | 0.9 | 93.3 |
| Gangxico | 5218 | 23.9 | 73.8 | 1.7 | 0.0 |
| Jialongco | 4374 | 81.2 | 11.5 | 7.3 | 76.5 |
| Longmuqieco | 5358 | 29.9 | 70.1 | 0.0 | 15.0 |