# Peer review of "Changes in glacial lakes in the Poiqu River Basin in the central Himalayas"

_Hydrology and Earth System Sciences, 2020_

## Short Comment (SC1) · 14 Nov 2020

I feel this manuscript is a very interesting research with abundance of first hand data. To my knowledge, there are so many papers on glaciers and glacial lakes in Himalayas, and most are glacier and glacier lake inventory and their changes on regional scale and provide only overall statistics (e.g., Nie et al., 2014, 2017, 2018), seldom one sees detailed descriptions for theier contribution on water balance in individual lakes and glaciers . What I am most interested in is the reconstruction of the temporal changes in several lakes and their water balance calculations, which is mainly absent in glacier lakes. Involved in these points are some details I think should be further clarified: 1) the identification of glaciers and glacial lakes, especially their quantities (e.g., sizes, areas, volumes), which are crucial for the variation assertion.  2) temperature and rainfall

data seem relatively weak in supporting the variation in glaciers and glacial lakes. No remarkable trend coincidence is established between them. The authors may consider the alternative possibilities that the effect of global warming might be little in such a small area. Then the variations may be more resulted from local conditions instead of global effects. This could be discussed in a broad scope. 3) the reconstruction of GB lake below the water level (i.e., Fig.18) is very helpful for tracing the lake changes, but the method in section 5.1 should be more detailed in briefing the technique and calculation. And it needs generalization to other lakes. 4) equations for water balance involve general realizations of glacier and snow melt water, which should be argued in more details with references. In other words, the parameters involved in Eq.6-11 are not well discussed one by one in details. Moreover, empirical parameters in relation to local conditions should be more specific to individual lakes and glaciers. 5) The water balance calculations are listed in tables, which are not well clear to reveal the correlation between water and lake variations. It is suggested that these results should be plotted with comparing curves. In summary, this work is a good contribution to the study of glaciers and glacial lakes in the central Himalayas; and its discussion on water balance can be generalized to the water resources in Tibetan Plateau.

---

## Short Comment (SC2) · 17 Nov 2020

Much attentions are paid on the dynamic of glaciers and glacial lakes in the process of recent climate change recently. Though many documents focused on the changes of glacier and glacial lake in Poiqu or Himalayas (including Poiqu basin), this submission outstood by more detailed analysis and gross reasonable water balance calculation. So, this is an original paper on an interesting topic of changes in glacial lakes in the Poiqu River Basin. However, the authors should be asked to correct some deficiencies in the paper prior to publication. In general, (1) the expression pattern of the manuscript is overall a little of bits and pieces. The authors provided many detailed changing stories of glaciers and glacial lakes, but it seems insufficient in generalization, particularly, for tables and figures. (2) I suggested that the authors reconsidered

the contents and structures of the manuscript (as it is not usually with the special sections of data, methods in the manuscript), and some of the contents in the section of results should be moved to the section of discussion. (3) The assessments of errors and uncertainties risen from the different data source and data processing should be more expressed in the manuscript. Some minor details comments: (1) Page 2, line 19-21: There is a new publication of about glacial lakes in Himalayas. Also, a few new documents can be referred to give the popular classification of glacial lake. (2) Page 2, line 26: Please provide the reference. (3) Page 3, line 14: Carefully summarized this expression: the retreat of glaciers and the growth of lakes are generally believed to be caused by decreasing rainfall. For it is difficult to draw a conclusion that the rainfall was widely decreasing. (4) Page 3, line 1: Is there any manual vectorization/revision performedïij§ (5) Page 3, Fig.4 and Table 3: Some technical terms, such as the type of glacial lake, are not professional. (6) In table 4, How did you get the position of glacial lake? For example, it is the geometry center position of glacial lake? The types of glacial lake is not consistent with that in the text. (7) I think the contents of Fig.6 and Table 7 repeat to some degree. (8) Page 9, line 8-11: It is belonged to discussion. (9) Page 9, line 21: "Speed" is incorrect expression here. (10) I think Table 8 can be changed into Figure. (11) Some like Fig. 7-10 can be presented by attachments. (12) Page 11, line 19: Data source? (13) Page 11, line 21-22: Data source? (14) Page 14, line 7: The hydraulically connection is not visible. (15) Page 14, line 10-12: It can be deleted. (16) Page 15, line 29: In some cases, the moraine dam which is frozen soil did not have high porosity. (16) Page 17, line 26, 31: Table 9 and 6 are Table 10? (17) Page 17, line 16-18: Also considering the frozen soil? (18) Page 17, line 16-21:Listing them as note in the bottom of Table 11.

---

## Author Comment (AC1) · 25 Nov 2020

Reply to the comments. We appreciate Dr. Shang guan for his instructive comments of our manuscript. At present we'd like provide a general reply to his questions. About comment 1, We consider as follows: 1). For the identification of glaciers and glacial lakes we have employed multi-source data and RS images , combining with UAV and field surveys ; and we can give more detailed descriptions of how the data are obtained in the revised version . 2). As it is well known, weather and climate data in Himalayas at watershed scale is rare and only records in near counties are sometimes available; but these data is insufficient to reflect the local changes. On the other hand, local fluctuation of temperature and rainfall (snowfall) do not necessarily coincide with the global changes. The suggestion of local variation rather than consequence of global warming

is interesting and worth further consideration. 3). GB reconstruction is crucial to water balance calculation; we'll give pay more attention to the technique and validity of the process. But this requires high resolution images of many years, which is always available in practice. For the reliability of the method we can make further clarification. 4). Actually, there are so many methods estimating to melt water and involving glacier dynamics. Detailed discussions about the related parameters are promised to be added in the revision. 5). We have plotted the curves for comparison between calculation and observation for water balance of the lakes; they are omitted in the text only for page save. The coincidence is very good in general, and should be added in the revised text.

---

## Author Comment (AC2) · 25 Nov 2020

Reply to the comments

We appreciate Dr. Wang for his instructive comments of our manuscript. At present we'd like provide a general reply to his questions.

As for comment 2, we note that Dr. Wang has proposed many good and instructive advices on the expression of the manuscript. General we agree and we'll consider each of the suggestions in the revision. The most important point highlighted here is concerning the errors and uncertainties of the data; and these can be clarified by making a more detailed description of the data source and data processing. In addition, the organization of the origin text may be reorganized in a more clear and logic way.

---

## Short Comment (SC3) · 10 Apr 2021

The Himalayas is one of the most sensitive regions to climate change, which present conspicuous changes in glaciers and glacial lakes. This manuscript is interesting in that it gives detailed annual variations of several lakes and reconstructs the history, which fills the gap of lacking case studies in the area. The major flaws are the data reliability and the parameter estimation in the construction of the water balance equation. For instance, the following should be further clarified.

1) The climate and hydrology in section 2.2 are irrelative to the topic because the description gives only the present situation, which is insufficient to reflect their long-term changes. The gross trend of both should be mentioned to some extent. 2) Section 3.3

says that Table 5 lists tributary parameters crucial for lake evolution, but one cannot see how they influence lake evolution. In general, what are the major factors influencing the glacial lakes, the climate or the local geomorphology? This is especially important when we find that the climate data present in the manuscript does not show any positive correlation to the changes in glacial lakes (i.e., section 4.2). 3) Section 4.1 shows three patterns of lake variations, which seem to be related, partly, to the elevation. As mentioned above, the effects due to local morphology and hydrology should be discussed as possible influential agencies. After all, there are hydrologically connected lakes, as mentioned in the text. 4) The relation between glaciers and glacial lakes is clearly revealed, but their relations to the climate data are not well identified in Fig.15 $\sim$ 17. This is related to the problem mentioned in 2). 5) The reconstruction of the lake in section 5 needs further clarification to guarantee the reliability of the volume estimation, which is the base for the WBE establishment. 6) Because so many parameters are involved in Eq.6-15, more details should be added to the estimation methods and accuracy. It would be better if related data were used for comparison, even in other areas. For example, for the infiltration, the text has cited empirical formula in literature, but it does not provide details about how these are really used for individual lakes. In short, all the estimations are not well-grounded. 7) There are 147 lakes mentioned in the study area, but only 5 lakes are studied in detail. Although it is perhaps supercritical to require such investigation to every lake, it is reasonable to ask for a general understanding of the lakes based on the case studies. However, the text seems to end with the several lakes, almost ignoring the overview of the lakes in the central Himalayas just as emphasized in the title. Therefore, it is suggested that there should be some discussions on this missing point.

---

## Author Comment (AC3) · 15 Apr 2021

Reply to the comment of Adhikari

Dear editor

We're pleased to see the comment from Dr. Adhikari and thanks for his opinions. Now we'd like to give a brief reply, and in the revision we'll take into account all the opinions.

1) Concerning the climate and hydrology background, we have made great attempt to collect the local data and made multi-factor monitoring of weather in Galong lake since last rainy season and collected the long-term climate data from Nielamu, these data are expected to be helpful for further understanding of the climate and weather impact

in the study area. 2) and 3) Geomorphology conditions can be easily formulated by GIS, and we'll make comparisons between these for different lakes in the study area. This will give a more clear picture of glacial lakes under different topographic conditions. And this may as well clarify the questions about hydrology influence. 4) We've got more data (e.g., from Nielamu) for the establishing relationships between climate and glaciers and the associated glacial lakes. 5) We have conducted underwater measurements in the lake of Galongcuo and Jialongcuo (two of the five examples studied in this paper) and other two small lakes, which will enable us to build more accurate correlation between the lake area and impounding volume. 6) Some parameters concerning the melt-water estimation are available now, such as the radiative index and granular texture of the moraines. Accuracy evaluation is possible for the water estimates. 7) Yes, it is important to give an overall view of the glacial lakes in the central Himalayas based on the case studies. As the lake distribution on regional scale is well known, the evolution trend of these lakes are relatively easy to evaluate. And this will reinforce the significance of the study.

---

## Referee Comment (RC1) · Anonymous Referee #1 · 14 May 2021

The authors present a study of the state and historical evolution of glacial lakes in the Poiqu River basin in the Himalayas. The subject is interesting to a wider public, mainly due to the speed of the on-going changes that take place at high altitudes in the Himalayas caused by climate warming.

The paper gives a thorough description of changes that take place in the region. There are 20 figures and 13 tables illustrating every aspect of that description. Unfortunately, the paper is rather disappointing. The description of changes that take place in the region, manifested by a decrease of glaciers and an increase of glacial lakes, is too long and too detailed, while a synthesis of the changes is missing. At present, the reader does not learn much new about the processes involved.

It is claimed that those changes depend on local conditions, with snow-melt being predominant source of lake water increase at high altitudes while the melting glaciers are mainly contributing at lower latitudes. The water balance equation (WBE) developed follows simple, standard assumptions. The only data processing is applied to derive annual changes in the lake water volumes based on historical satellite images from the time period (1977-2016) and the available elevation data.

The authors compare the WBE model water volume estimates with observed water volume changes; however the accuracy of the modelled and observed values is not given. Therefore we do not know what is the predictive power of the WBE model. It would be also interesting to learn how the approach described can be generalised to other case studies.

---

## Referee Comment (RC2) · Anonymous Referee #2 · 24 Jun 2021

The manuscript is interesting to read and will attract a wider scientific audience since the impact of climate change on the Himalayas region is a big concern worldwide. Moreover, this manuscript provides some important information about the current state and historical changes of glacial lakes in the study area. That being said, this paper is within the scope of HESS and will likely be a significant contribution and can be recommended for publication, although, some minor changes and a few clarifications should be made.

Some parts of the manuscript seem to be too long and detailed, and, yet, some parts of the text need either more information or further clarifications about the processes (this will allow some better understanding).

The scientific quality of this paper is overall good, but it feels like authors could provide more information about the WBE. My overall impression is that the authors used the available data/historical information and I do not see any see any methodological issues with their approach. However, the reliability of their proposed method/approach should be discussed a bit more. I am not suggesting that authors are not aware of this. I am certain they are aware just feel that a few more sentences discussing this could help the paper and could provide valuable information to readers (although, some parts in the Discussion section are very well written and cover certain aspects).

The presentation of the data/results is excellent (all figures and tables are produced to a high standard and fit well within the text).

Here are just a few minor advices to the authors:

It would be good if authors could go through their introduction again and clarify further a few things. For example, page 2, lines 14 and 15: "The reduction in the south is much larger than that in the north (Wei et al., 2014)" could follow a short explanation why is that, despite the fact it is obvious and/or could be found in the cited paper, so readers can understand better the context while they're reading.

Also, it seems that references are missing in some parts. For example, page 2, lines 25 and 26: "Statistics show that the expansion accounts for 67% of the area increase, while the formation of a new glacial lake contributes only 33%." It's unclear if this statistics can be found in Wang et al., 2015 that is cited a little bit later or not? Please clarify and add references where such confusion could arise.

Some sentences could be rephrased. For example, page 14, lines 16 and 17: "To understand changes in glacial lakes, it is necessary to find the changes in water volume in the lakes. Then, we must find the lake volume from the area." can be formulated something like this: To understand changes in glacial lakes, it is crucial to find the differences in water volume in the lakes. In the next step, it is necessary to find the lake volume from the area. (This is only a suggestion to authors).
Importantly, page 16 line 36 and page 17 line 1: "Physical models have incorporated many influencing factors, such as temperature and radiation intensity; thus, these models have high calculation accuracy. However, they do not apply to areas lacking a sufficient database, as in the case in the Himalayas..." - I believe authors should more clearly highlight this somewhere in the beginning of the manuscript as it is important to state the lack of data from this region (which is a fact) and how such lack of data influence scientific decisions in terms of study design immediately to readers.

---

## Author Comment (AC4) · 21 Jul 2021

We appreciate Anonymous Referee #1 for his or her instructive comments of our manuscript. At present we'd like provide a general reply to these questions.

Anonymous Referee #1 The authors present a study of the state and historical evolution of glacial lakes in the Poiqu River basin in the Himalayas. The subject is interesting to a wider public, mainly due to the speed of the on-going changes that take place at high altitudes in the Himalayas caused by climate warming. The paper gives a thorough description of changes that take place in the region. There are 20 figures and 13 tables illustrating every aspect of that description. Unfortunately, the paper is rather disappointing. The description of changes that take

place in the region, manifested by a decrease of glaciers and an increase of glacial lakes, is too long and too detailed, while a synthesis of the changes is missing. At present, the reader does not learn much new about the processes involved.

Reply: Thank you very much for the comments. Indeed, in the original manuscript, we gave too much detailed description of glaciers and glacial lakes, especially glacial lakes, and ignored the discussion and analysis of multi-year change trend. The description of the overall changes of glaciers and glacial lakes is not comprehensive and in-depth in section 4 of "variations in glaciers and glacial lakes". After submission of the original manuscript, we have collected additional data of the local geological and geomorphic backgrounds, which will be incorporated in the revision text to give a better and comprehensive picture of the glacial lake changes at regional scale. We will rewrite section 4 to include new discussions on the chapter 4.1 "the distribution and trend of glaciers and glacial lakes", the chapter 4.2 "the development and change of typical glaciers and glacial lakes in the study area, and the causes and trends of the changes from history to future. At the same time, we will delete some tedious details of the 5 typical glacial lakes to improve the readability.

It is claimed that those changes depend on local conditions, with snow-melt being predominant source of lake water increase at high altitudes while the melting glaciers are mainly contributing at lower latitudes. The water balance equation (WBE) developed follows simple, standard assumptions. The only data processing is applied to derive annual changes in the lake water volumes based on historical satellite images from the time period (1977-2016) and the available elevation data.

Reply: In the original manuscript, we did not explain the data clearly enough in the chapter 3 "Data sources". And we mainly introduce the source and accuracy of the image data. In the later revision, we plan to add a chapter 3.1.2 to introduce the source and accuracy of the image data and meteorological data. All the data, which involved mainly include the relevant meteorological data in WBE, will be introduced in detail. The calculation formula WBE is more suitable for the regions with a paucity of data.

The WBE ( V = P + G -I -E) follows a simple standard assumption, but parameters, calculation of P, G, I and E are the selected formulas for the regions with a paucity of data.

The authors compare the WBE model water volume estimates with observed water volume changes; however the accuracy of the modelled and observed values is not given. Therefore we do not know what the predictive power of the WBE model is. It would be also interesting to learn how the approach described can be generalized to other case studies

Reply: In the original manuscript, we have introduced the data sources and accuracy of the image but not explained the data clearly enough. In the later revision, we plan to add a section to explain the dominated processes in glacial lake changes and reinforce our assumptions for the WBE. All the involved data in the WBE will be specified in details. In the present form, the WBE V = P + G -I -E involve the formula as follow: PR,PS,G, I and E. The above formula contains the following parameters: 1. The geomorphic parameters are achieved by satellite image, including drainage area contributing to the lake (S), the geomorphologic factors such as slope and lake area (A). 2. The meteorological data is achieved from National Meteorological Data Center (http://data.cma.cn/), including rainfall intensity (R), snowfall (PS), rainfall quantity (QR), temperature (T), solar radiation (IR), and wind speed (v). 3. Parameters are measured by field and indoor experiments, including snow density ( S), snow permeability (K) and glacier density ( G). 4. The parameters are obtained by querying the past information, including fracture density , surface saturated vapor pressure (p) and turbulent energy

In the later revision, we plan to describe the approach how to calculate WBE more clearly, and give a more exhaustive comprehensive calculation formula, discuss the reliability and the possible error. It may be useful to generalize to other case studies.

Please also note the supplement to this comment:

https://hess.copernicus.org/preprints/hess-2020-20/hess-2020-20-AC4-supplement.pdf

---

## Author Comment (AC5) · 21 Jul 2021

We appreciate Anonymous Referee #2 for his or her instructive comments of our manuscript. At present we'd like provide a general reply to these questions.
and can be recommended for publication, although, some minor changes and a few clarifications should be made.

Some parts of the manuscript seem to be too long and detailed, and, yet, some parts of the text need either more information or further clarifications about the processes (this will allow some better understanding). The scientific quality of this paper is overall good, but it feels like authors could provide more information about the WBE. My overall impression is that the authors used the available data/historical information and I do not see any see any methodological issues with their approach. However, the reliability of their proposed method/approach should be discussed a bit more. I am not suggesting that authors are not aware of this.

I am certain they are aware just feel that a few more sentences discussing this could help the paper and could provide valuable information to readers (although, some parts in the Discussion section are very well written and cover certain aspects).

Reply: Both reviewers have proposed the same advice. In the original manuscript, we did not explain the WBE model clearly enough. So in the later revision, we plan to describe the approach of calculating the WBE more clearly, and give an exhaustive formula, discuss the reliability and the possible error. It may be helpful to generalize to other case studies.

The presentation of the data/results is excellent (all figures and tables are produced to a high standard and fit well within the text).

Here are just a few minor advices to the authors:

It would be good if authors could go through their introduction again and clarify further a few things. For example, page 2, lines 14 and 15: "The reduction in the south is much larger than that in the north (Wei et al., 2014)" could follow a short explanation why is that, despite the fact it is obvious and/or could be found in the cited paper, so readers can understand better the context while they're reading.

Reply: We will check and modify the logic of the manuscript to enhance readability. It is good for readers to understand better the context. This is our ongoing work. In the revised manuscript, we will revise 1 introduction, especially line 15-17 in the page 2, line 3-12 in the page 3, and added the research of WBE literature.

Also, it seems that references are missing in some parts. For example, page 2, lines 25 and 26: "Statistics show that the expansion accounts for 67% of the area increase, while the formation of a new glacial lake contributes only 33%." It's unclear if this statistics can be found in Wang et al., 2015 that is cited a little bit later or not? Please clarify and add references where such confusion could arise. Some sentences could be rephrased. For example, page 14, lines 16 and 17: "To understand changes in glacial lakes, it is necessary to find the changes in water volume in the lakes. Then, we must find the lake volume from the area." can be formulated something like this: To understand changes in glacial lakes, it is crucial to find the differences in water volume in the lakes. In the next step, it is necessary to find the lake volume from the area. (This is only a suggestion to authors).

Importantly, page 16 line 36 and page 17 line 1: "Physical models have incorporated many influencing factors, such as temperature and radiation intensity; thus, these models have high calculation accuracy. However, they do not apply to areas lacking a sufficient database, as in the case in the Himalayas. . ." - I believe authors should more clearly highlight this somewhere in the beginning of the manuscript as it is important to state the lack of data from this region (which is a fact) and how such lack of data influence scientific decisions in terms of study design immediately to readers.

Reply: Thank the reviewer so much for giving us many useful suggests. We plan to modify our sentences throughout the manuscript. The chapter 2 "study area" will be highlighted that the region is the absence of data in the study area background presentation. And we also consider to add a chapter 3.1.2 to introduce the source and accuracy of the image data and meteorological data. At the same time, we will supplement the literature to review of relevant models in section 1 "Instruction" and
discuss defects of existing models and methods. In the section 6 "Discussions", we will discuss the characteristics and accuracy of our WBE method.

Please also note the supplement to this comment:
https://hess.copernicus.org/preprints/hess-2020-20/hess-2020-20-AC5-supplement.pdf

―――――――――――――――――――

---

## Author Response (AR1)

The authors present a study of the state and historical evolution of glacial lakes in the Poiqu River basin in the Himalayas. The subject is interesting to a wider public, mainly due to the speed of the on-going changes that take place at high altitudes in the Himalayas caused by climate warming.

The paper gives a thorough description of changes that take place in the region. There are 20 figures and 13 tables illustrating every aspect of that description. Unfortunately, the paper is rather disappointing. The description of changes that take place in the region, manifested by a decrease of glaciers and an increase of glacial lakes, is too long and too detailed, while a synthesis of the changes is missing. At present, the reader does not learn much new about the processes involved.

It is claimed that those changes depend on local conditions, with snow-melt being predominant source of lake water increase at high altitudes while the melting glaciers are mainly contributing at lower latitudes. The water balance equation (WBE) developed follows simple, standard assumptions. The only data processing is applied to derive annual changes in the lake water volumes based on historical satellite images from the time period (1977-2016) and the available elevation data.

The authors compare the WBE model water volume estimates with observed water volume changes; however the accuracy of the modelled and observed values is not given. Therefore we do not know what the predictive power of the WBE model is. It would be also interesting to learn how the approach described can be generalized to other case studies.

**Reply:**

We appreciate Anonymous Referee #1 for his or her instructive comments of our manuscript. At present we'd like provide a general reply to these questions.

1)The authors present a study of the state and historical evolution of glacial lakes in the Poiqu River basin in the Himalayas. The subject is interesting to a wider public, mainly due to the speed of the on-going changes that take place at high altitudes in the Himalayas caused by climate warming.

The paper gives a thorough description of changes that take place in the region. There are 20 figures and 13 tables illustrating every aspect of that description. Unfortunately, the paper is rather disappointing. The description of changes that take place in the region, manifested by a decrease of glaciers and an increase of glacial lakes, is too long and too detailed, while a synthesis of the changes is missing. At present, the reader does not learn much new about the processes involved.

**Reply:** We've added section 2.3 ("Distribution of glacial lakes in the Poiqu River Basin") to illustrate the overall state of glacier and glacial lake distribution in the study area. At the same time, we have deleted 4 figures and replaced 4 tables and rearranged the e materials to give a clearer picture of the changes in glacial lakes and the related glaciers.

2)It is claimed that those changes depend on local conditions, with snow-melt being predominant source of lake water increase at high altitudes while the melting glaciers are mainly contributing at lower latitudes. The water balance equation (WBE) developed follows simple, standard assumptions. The only data processing is applied to derive annual changes in the lake water volumes based on historical satellite images from the time period (1977-2016) and the available elevation data.

**Reply:** To our knowledge there is no study on the changes in individual glacial lakes in the Himalayas, and we make the first attempt to give a framework of describing the evolution by calculating the water variations. As data available are much limited, we can only give the annual changes. In section 2.2 we have set 4 subsections to illustrate the data sources and processing and their accuracy. All the data involved in WBE are introduced in details.

The calculated results of the five lakes indicate that, although the WBE is based on a simple standard assumption, it works well for the regions with a paucity of data and reveals the fundamental facts ignored in a study on the regional scale. Following the routine in our paper, this method is expected to be more effective when the involved data (especially the conditions of local weather and glaciers) are sufficiently available.

3)The authors compare the WBE model water volume estimates with observed water volume changes; however the accuracy of the modelled and observed values is not given. Therefore we do not know what the predictive power of the WBE model is. It would be also interesting to learn how the approach described can be generalized to other case studies

**Reply:** In the revised revision, we have added a section to explain the dominated processes in glacial lake changes and reinforce our assumptions for the WBE. In particular, we give a detailed description of water volume calculation of the lake, which guarantees the reliability of the real volume on field survey level. Then we give a detailed formula of WBE calculation in practice and list the errors in comparison with the real values in Table 12. This may establish the practical applicability of the WBE method.

Interactive comment on "Changes in glacial lakes in the Poiqu River Basin in

the central Himalayas" by Pengcheng Su et al.

**Anonymous Referee #2**

The manuscript is interesting to read and will attract a wider scientific audience since the impact of climate change on the Himalayas region is a big concern worldwide.

Moreover, this manuscript provides some important information about the current state and historical changes of glacial lakes in the study area. That being said, this paper is within the scope of HESS and will likely be a significant contribution and can be recommended for publication, although, some minor changes and a few clarifications should be made.

Some parts of the manuscript seem to be too long and detailed, and, yet, some parts of the text need either more information or further clarifications about the processes (this will allow some better understanding).

The scientific quality of this paper is overall good, but it feels like authors could provide more information about the WBE. My overall impression is that the authors used the available data/historical information and I do not see any see any methodological issues with their approach. However, the reliability of their proposed method/approach should be discussed a bit more. I am not suggesting that authors are not aware of this.

I am certain they are aware just feel that a few more sentences discussing this could help the paper and could provide valuable information to readers (although, some parts in the Discussion section are very well written and cover certain aspects).

The presentation of the data/results is excellent (all figures and tables are produced to a high standard and fit well within the text).

Here are just a few minor advices to the authors:

It would be good if authors could go through their introduction again and

clarify further a few things. For example, page 2, lines 14 and 15: "The reduction in the south is much larger than that in the north (Wei et al., 2014)" could follow a short explanation why is that, despite the fact it is obvious and/or could be found in the cited paper, so readers can understand better the context while they're reading.

Also, it seems that references are missing in some parts. For example, page 2, lines 25 and 26: "Statistics show that the expansion accounts for 67% of the area increase, while the formation of a new glacial lake contributes only 33%." It's unclear if this statistics can be found in Wang et al., 2015 that is cited a little bit later or not? Please clarify and add references where such confusion could arise. Some sentences could be rephrased. For example, page 14, lines 16 and 17: "To understand changes in glacial lakes, it is necessary to find the changes in water volume in the lakes. Then, we must find the lake volume from the area." can be formulated something like this: To understand changes in glacial lakes, it is crucial to find the differences in water volume in the lakes. In the next step, it is necessary to find the lake volume from the area. (This is only a suggestion to authors).

Importantly, page 16 line 36 and page 17 line 1: "Physical models have incorporated many influencing factors, such as temperature and radiation intensity; thus, these models have high calculation accuracy. However, they do not apply to areas lacking a sufficient database, as in the case in the Himalayas. . ." - I believe authors should more clearly highlight this somewhere in the beginning of the manuscript as it is important to state the lack of data from this region (which is a fact) and how such lack of data influence scientific decisions in terms of study design immediately to readers.

**Reply:**

We appreciate Anonymous Referee #2 for his or her instructive comments of our manuscript. At present we'd like provide a general reply to these questions.

The manuscript is interesting to read and will attract a wider scientific audience since the impact of climate change on the Himalayas region is a big concern worldwide.

Moreover, this manuscript provides some important information about the current state and historical changes of glacial lakes in the study area. That being said, this paper is within the scope of HESS and will likely be a significant contribution and can be recommended for publication, although, some minor changes and a few clarifications should be made.

Some parts of the manuscript seem to be too long and detailed, and, yet, some parts of the text need either more information or further clarifications about the processes (this will allow some better understanding).

The scientific quality of this paper is overall good, but it feels like authors could provide more information about the WBE. My overall impression is that the authors used the available data/historical information and I do not see any see any methodological issues with their approach. However, the reliability of their proposed method/approach should be discussed a bit more. I am not suggesting that authors are not aware of this.

I am certain they are aware just feel that a few more sentences discussing this could help the paper and could provide valuable information to readers (although, some parts in the Discussion section are very well written and cover certain aspects).

The presentation of the data/results is excellent (all figures and tables are produced to a high standard and fit well within the text).

**Reply:** We've checked and modified the logic in the revision. Some details of

glacial lakes are deleted, including related tables and figures (e.g. Fig.1, Fig.7, Fig.12, Fig.13, Fig.17, Table 1, Table 2, Table 3, Table 4, Table 12, and Table 13.). The abstract and conclusions have been rewritten.

Here are just a few minor advices to the authors:

It would be good if authors could go through their introduction again and clarify further a few things. For example, page 2, lines 14 and 15: "The reduction in the south is much larger than that in the north (Wei et al., 2014)" could follow a short explanation why is that, despite the fact it is obvious and/or could be found in the cited paper, so readers can understand better the context while they're reading.

**Reply:** All these points have been considered or revised in the new revised manuscript.

Also, it seems that references are missing in some parts. For example, page 2, lines 25 and 26: "Statistics show that the expansion accounts for 67% of the area increase, while the formation of a new glacial lake contributes only 33%." It's unclear if this statistics can be found in Wang et al., 2015 that is cited a little bit later or not? Please clarify and add references where such confusion could arise. Some sentences could be rephrased. For example, page 14, lines 16 and 17: "To understand changes in glacial lakes, it is necessary to find the changes in water volume in the lakes. Then, we must find the lake volume from the area." can be formulated something like this: To understand changes in glacial lakes, it is crucial to find the differences in water volume in the lakes. In the next step, it is necessary to find the lake volume from the area. (This is only a suggestion to authors).

**Reply:** All these points have been considered or revised in the new revised manuscript.

Importantly, page 16 line 36 and page 17 line 1: "Physical models have

incorporated many influencing factors, such as temperature and radiation intensity; thus, these models have high calculation accuracy. However, they do not apply to areas lacking a sufficient database, as in the case in the Himalayas. . .” - I believe authors should more clearly highlight this somewhere in the beginning of the manuscript as it is important to state the lack of data from this region (which is a fact) and how such lack of data influence scientific decisions in terms of study design immediately to readers.

**Reply:** We revised our sentences throughout the manuscript. The chapter 2 "Background of the Poiqu River Basin and Data sources" is highlighted that the region is the absence of data in the study area background presentation. And we also added a chapter 2.2.3 Sources of meteorological data and 2.2.4 Processing method of meteorological data to introduce the source and accuracy of the image data and meteorological data. In the section 4.4.2 Water balance for 5 typical lakes, we discuss the characteristics and accuracy of our WBE method. Table 12 lists the calculated values, measured values and errors of the volumes of five typical ice lakes.
I feel this manuscript is a very interesting research with abundance of first hand data. To my knowledge, there are so many papers on glaciers and glacial lakes in Himalayas, and most are glacier and glacier lake inventory and their changes on regional scale and provide only overall statistics (e.g., Nie et al., 2014, 2017, 2018), seldom one sees detailed descriptions for their contribution on water balance in individual lakes and glaciers. What I am most interested in is the reconstruction of the temporal changes in several lakes and their water balance calculations, which is mainly absent in glacier lakes. Involved in these points are some details I think should be further clarified: 1) the identification of glaciers and glacial lakes, especially their quantities (e.g., sizes, areas, volumes), which are crucial for the variation assertion. 2) temperature and rainfall data seem relatively weak in supporting the variation in glaciers and glacial lakes. No remarkable trend coincidence is established between them. The authors may consider the alternative possibilities that the effect of global warming might be little in such a small area. Then the variations may be more resulted from local conditions instead of global effects. This could be discussed in a broad scope. 3) the reconstruction of GB lake below the water level (i.e., Fig.18) is very helpful for tracing the lake changes, but the method in section 5.1 should be more detailed in briefing the technique and calculation. And it needs generalization to other lakes. 4) equations for water balance involve general realizations of glacier and snow melt water, which should be argued in more details with references. In other words, the parameters involved in Eq.6-11 are not well discussed one by one in details. Moreover,

empirical parameters in relation to local conditions should be more specific to individual lakes and glaciers. 5) The water balance calculations are listed in tables, which are not well clear to reveal the correlation between water and lake variations. It is suggested that these results should be plotted with comparing curves. In summary, this work is a good contribution to the study of glaciers and glacial lakes in the central Himalayas; and its discussion on water balance can be generalized to the water resources in Tibetan Plateau.

**Reply:**

We appreciate Dr. Shang guan for his instructive comments of our manuscript. At present we'd like to provide a general reply to his/her questions.

1) the identification of glaciers and glacial lakes, especially their quantities (e.g., sizes, areas, volumes), which are crucial for the variation assertion.

Reply: We added chapter 2.3 "distribution of glacial lakes in the poiqu River Basin", discussed area distribution, altitude distribution and types of glacial lakes in the poiqu River Basin, and selected five typical glacial lakes (A>0.3 km$^2$) to explore the changes of glacial lakes and subsequent glaciers for many years.

2) temperature and rainfall data seem relatively weak in supporting the variation in glaciers and glacial lakes. No remarkable trend coincidence is established between them. The authors may consider the alternative possibilities that the effect of global warming might be little in such a small area. Then the variations may be more resulted from local conditions instead of global effects. This could be discussed in a broad scope.

Reply: Actually we have noted this problem, which in part might be ascribed to the lack of local data; and, perhaps more importantly, the temperature changes over a large scale do not necessarily impact the individual lakes and glaciers in local conditions in a small region as the study area. Nevertheless, the five typical lakes provide exemplification of the related changes, as discussed in Section 3.3 in the revision.

3) the reconstruction of GB lake below the water level (i.e., Fig.18) is very helpful for tracing the lake changes, but the method in section 5.1 should be more detailed in briefing the technique and calculation. And it needs generalization to other lakes.

Reply: We have added a section (4.1) to give a detailed description of the reconstruction of Lake Basin topography and the volume calculation. The procedures are as follows: 1) Interpret the water level at the lake boundary; 2) Transform the water-level vector data to point data in ArcGIS, with high point density representing high accuracy; 3) Assign DEM data to the point data. Then the average of the point data is the altitude of water-level (lake boundary).

4) equations for water balance involve general realizations of glacier and snow melt water, which should be argued in more details with references. In other words, the parameters involved in Eq.6-11 are not well discussed one by one in details. Moreover, empirical parameters in relation to local conditions should be more specific to individual lakes and glaciers.

Reply: We've made some discussions of the parameters and given the working formula for calculation (Eq.19 in the revision):

$$\Delta V = \alpha SR_a + R_{CS}\text{DDF}_S \cdot \text{PDD}_S \cdot S + R_{CG}\text{DDF}_G \cdot \text{PDD}_G \cdot A_G - KJAT$$

5) The water balance calculations are listed in tables, which are not well clear to reveal the correlation between water and lake variations. It is suggested that these results should be plotted with comparing curves. In summary, this work is a good contribution to the study of glaciers and glacial lakes in the central Himalayas; and its discussion on water balance can be generalized to the water resources in Tibetan Plateau.

Reply: We've tabulated the calculated results, measured results and the errors in Table 12. We tried to draw a picture. It is messy to draw a picture, includes all the 5 typical glacial lakes. But if we draw a single picture of a single glacial lake, five pictures will be added. At the same time, we think the error has been clearly seen in the table, so there is no figure, shows comparing curves. In the discussions, we do generalize the WBE to the evaluation of water sources in the Himalayas.
Much attentions are paid on the dynamic of glaciers and glacial lakes in the process of recent climate change recently. Though many documents focused on the changes of glacier and glacial lake in Poiqu or Himalayas (including Poiqu basin), this submission outstood by more detailed analysis and gross reasonable water balance calculation. So, this is an original paper on an interesting topic of changes in glacial lakes in the Poiqu River Basin. However, the authors should be asked to correct some deficiencies in the paper prior to publication. In general, (1) the expression pattern of the manuscript is overall a little of bits and pieces. The authors provided many detailed changing stories of glaciers and glacial lakes, but it seems insufficient in generalization, particularly, for tables and figures. (2) I suggested that the authors reconsidered the contents and structures of the manuscript (as it is not usually with the special sections of data, methods in the manuscript), and some of the contents in the section of results should be moved to the section of discussion. (3) The assessments of errors and uncertainties risen from the different data source and data processing should be more expressed in the manuscript. Some minor details comments: (1) Page 2, line 19-21: There is a new publication of about glacial lakes in Himalayas. Also, a few new documents can be referred to give the popular classification of glacial lake. (2) Page 2, line 26: Please provide the reference. (3) Page 3, line 14: Carefully summarized this expression: the retreat of glaciers and the growth of lakes are generally believed to be caused by decreasing rainfall. For it is difficult to draw a conclusion that the rainfall was widely decreasing. (4) Page 3, line 1: Is there

any manual vectorization/revision performed (5) Page 3, Fig.4 and Table 3: Some technical terms, such as the type of glacial lake, are not professional. (6) In table 4, How did you get the position of glacial lake? For example, it is the geometry center position of glacial lake? The types of glacial lake is not consistent with that in the text. (7) I think the contents of Fig.6 and Table 7 repeat to some degree. (8) Page 9, line 8-11: It is belonged to discussion. (9) Page 9, line 21: "Speed" is incorrect expression here. (10) I think Table 8 can be changed into Figure. (11) Some like Fig. 7-10 can be presented by attachments. (12) Page 11, line 19: Data source? (13) Page 11, line 21-22: Data source? (14) Page 14, line 7: The hydraulically connection is not visible. (15) Page 14, line 10-12: It can be deleted. (16) Page 15, line 29: In some cases, the moraine dam which is frozen soil did not have high porosity. (16) Page 17, line 26, 31: Table 9 and 6 are Table 10? (17) Page 17, line 16-18: Also considering the frozen soil? (18) Page 17, line 16-21: Listing them as note in the bottom of Table 11.

**Reply :**

We appreciate Dr. Wang for his instructive comments of our manuscript. At present we'd like to provide a general reply to his/her questions.

In general, (1) the expression pattern of the manuscript is overall a little of bits and pieces. The authors provided many detailed changing stories of glaciers and glacial lakes, but it seems insufficient in generalization, particularly, for tables and figures.

Reply: As a reply to the first comment, we've deleted some details and figures and tables and rearranged the materials for more readability.

(2) I suggested that the authors reconsidered the contents and structures of the manuscript (as it is not usually with the special sections of data, methods in the manuscript), and some of the contents in the section of results should be

moved to the section of discussion.

Reply: Yes, we've re-organized the materials and rewritten the conclusions and discussions.

(3) The assessments of errors and uncertainties risen from the different data source and data processing should be more expressed in the manuscript.

Reply: We add a section (2.2) to express the data sources and processing method.

Some minor details comments:

(1) Page 2, line 19-21: There is a new publication of about glacial lakes in Himalayas. Also, a few new documents can be referred to give the popular classification of glacial lake.

(2) Page 2, line 26: Please provide the reference.

Reply: New references are added into the revised version.

(3) Page 3, line 14: Carefully summarized this expression: the retreat of glaciers and the growth of lakes are generally believed to be caused by decreasing rainfall. For it is difficult to draw a conclusion that the rainfall was widely decreasing.

Reply: This sentence was revised.

(4) Page 3, line 1: Is there any manual vectorization/revision performed

Reply: This sentence was revised.

(5) Page 3, Fig.4 and Table 3: Some technical terms, such as the type of glacial lake, are not professional.

(6) In table 4, How did you get the position of glacial lake? For example, it is the geometry center position of glacial lake? The types of glacial lake is not consistent with that in the text.

Reply: We delete the old table 2 and changed Fig.4, Table 3 and Table 4.

(7) I think the contents of Fig.6 and Table 7 repeat to some degree.

Reply: The figures are revised to avoid the repetition.

(8) Page 9, line 8-11: It is belonged to discussion. (9) Page 9, line 21: "Speed"

is incorrect expression here.

Reply: We've changed the speed as "annual change rate".

(10) I think Table 8 can be changed into Figure.

(11) Some like Fig. 7-10 can be presented by attachments.

(12) Page 11, line 19: Data source? (13) Page 11, line 21-22: Data source?

(14) Page 14, line 7: The hydraulically connection is not visible.

(15) Page 14, line 10-12: It can be deleted.

Reply: All these points have been considered or revised in the new text. We've changed some tables and figures for clarity.

(16) Page 15, line 29: In some cases, the moraine dam which is frozen soil did not have high porosity.

Reply: In the context we mainly consider the permeability of the substrate of the lake and channel, but not the end moraine dam.

(16) Page 17, line 26, 31: Table 9 and 6 are Table 10?

(17) Page 17, line 16-18: Also considering the frozen soil?

(18) Page 17, line of 16-21: Listing them as note in the bottom of Table 11.

Reply: We have incorporated all these opinions in the revision, and added note to Table 11.
The Himalayas is one of the most sensitive regions to climate change, which present conspicuous changes in glaciers and glacial lakes. This manuscript is interesting in that it gives detailed annual variations of several lakes and reconstructs the history, which fills the gap of lacking case studies in the area. The major flaws are the data reliability and the parameter estimation in the construction of the water balance equation. For instance, the following should be further clarified.

1) The climate and hydrology in section 2.2 are irrelative to the topic because the description gives only the present situation, which is insufficient to reflect their long-term changes. The gross trend of both should be mentioned to some extent.

2) Section 3.3 says that Table 5 lists tributary parameters crucial for lake evolution, but one cannot see how they influence lake evolution. In general, what are the major factors influencing the glacial lakes, the climate or the local geomorphology? This is especially important when we find that the climate data present in the manuscript does not show any positive correlation to the changes in glacial lakes (i.e., section 4.2).

3) Section 4.1 shows three patterns of lake variations, which seem to be related, partly, to the elevation. As mentioned above, the effects due to local morphology and hydrology should be discussed as possible influential agencies. After all, there are hydrologically connected lakes, as mentioned in the text.

4) The relation between glaciers and glacial lakes is clearly revealed, but their relations to the climate data are not well identified in Fig.15~ 17. This is related to the problem mentioned in 2).

5) The reconstruction of the lake\ in section 5 needs further clarification to

guarantee the reliability of the volume estimation, which is the base for the WBE establishment.

6) Because so many parameters are involved in Eq.6-15, more details should be added to the estimation methods and accuracy. It would be better if related data were used for comparison, even in other areas. For example, for the infiltration, the text has cited empirical formula in literature, but it does not provide details about how these are really used for individual lakes. In short, all the estimations are not well-grounded.

7) There are 147 lakes mentioned in the study area, but only 5 lakes are studied in detail. Although it is perhaps supercritical to require such investigation to every lake, it is reasonable to ask for a general understanding of the lakes based on the case studies. However, the text seems to end with the several lakes, almost ignoring the overview of the lakes in the central Himalayas just as emphasized in the title. Therefore, it is suggested that there should be some discussions on this missing point.

**Reply:**

We're pleased to see the comment from Dr. Adhikari and thanks for his opinions. Now we'd like to give a brief reply, and in the revision we'll take into account all the opinions.

1) The climate and hydrology in section 2.2 are irrelative to the topic because the description gives only the present situation, which is insufficient to reflect their long-term changes. The gross trend of both should be mentioned to some extent.

Reply: The original Fig.3 is deleted in the revised version, and the original chapter 2.2 is rewritten.

2) Section 3.3 says that Table 5 lists tributary parameters crucial for lake evolution, but one cannot see how they influence lake evolution. In general, what are the major factors influencing the glacial lakes, the climate or the local geomorphology? This is especially important when we find that the climate

data present in the manuscript does not show any positive correlation to the changes in glacial lakes (i.e., section 4.2).

  3) Section 4.1 shows three patterns of lake variations, which seem to be related, partly, to the elevation. As mentioned above, the effects due to local morphology and hydrology should be discussed as possible influential agencies. After all, there are hydrologically connected lakes, as mentioned in the text.

4) The relation between glaciers and glacial lakes is clearly revealed, but their relations to the climate data are not well identified in Fig.15~ 17. This is related to the problem mentioned in 2).

Reply: We've comprehensively incorporated these opinions into the revised text. In particular, we have rewritten section 2 about the background of the Poiqu River Basin and the glaciers and glacial lakes, as well as the data sources.

5) The reconstruction of the lake\ in section 5 needs further clarification to guarantee the reliability of the volume estimation, which is the base for the WBE establishment.

Reply: We've added a section to describe the method, see reply to the previous comment. we rewrite chapter 5.1 to describe the calculation process and method of glacial lake volume in more detail, and add the description of calculation steps: the procedures are as follows: 1) interpret the water level as the lake boundary; 2) Transform the water-level vector data to point data in ArcGIS, with high point density representing high accuracy; 3) Assign DEM data to the point data. Then the average of the point data is the altitude of water level (lake boundary).

6) Because so many parameters are involved in Eq.6-15, more details should be added to the estimation methods and accuracy. It would be better if related data were used for comparison, even in other areas. For example, for the infiltration, the text has cited empirical formula in literature, but it does not provide details about how these are really used for individual lakes. In short, all

the estimations are not well-grounded.

Reply: We have thoroughly revised the text of the practical operation of the balance equation, providing discussions of the parameters and giving a working formula (Eq.19).

7) There are 147 lakes mentioned in the study area, but only 5 lakes are studied in detail. Although it is perhaps supercritical to require such investigation to every lake, it is reasonable to ask for a general understanding of the lakes based on the case studies. However, the text seems to end with the several lakes, almost ignoring the overview of the lakes in the central Himalayas just as emphasized in the title. Therefore, it is suggested that there should be some discussions on this missing point.

Reply: We have added section 2.3 for the overall picture of glacial lake distribution in the study area.